# An $AdS_3$ Dual for Supersymmetric MHV Celestial Amplitudes

Igor Mol

*State University of Campinas (Unicamp)**

We propose a generalisation of the Wess-Zumino-Novikov-Witten (WZNW) model, formulated on a holomorphic extension of supersymmetric three-dimensional Anti-de Sitter ($AdS_3$) space, which holographically reproduces the tree-level maximally-helicity-violating (MHV) celestial amplitudes for gravitons in $\mathcal{N} = 8$ supergravity and gluons in four-dimensional $\mathcal{N} = 4$ supersymmetric Yang-Mills (SYM) theory.

---

* igormol@ime.unicamp.br

**CONTENTS**

# I.  INTRODUCTION

This study represents the confluence of two distinct lines of inquiry. The first extends the investigations initiated by Ogawa *et al.* [1] and subsequently refined by Mol [2] into the correspondence between celestial conformal field theory (CFT) and (Euclidean) $AdS_3$ string theory. The holographic dictionary proposed in Mol [2] exhibited a structural asymmetry in the formulation of celestial vertex operators for gluons and gravitons within the celestial CFT. Specifically, while celestial gluon vertex operators were derived exclusively from solutions to the $AdS_3$ string theory equations of motion (whether from worldsheet primary fields or boundary current algebras on $AdS_3$) the graviton vertex operators relied on an auxiliary set of operators defined on the celestial sphere. These additional operators effectively "dressed" the graviton vertex operators to ensure that their correlators reproduced the Berends-Giele-Kuijf (BGK) formula describing the tree-level maximally-helicity-violating (MHV) scattering amplitudes for gravitons in Einstein's gravity.

In the present work, we derive an alternative representation for celestial graviton leaf amplitudes, employing an operator factorisation approach that will be detailed below. It will be demonstrated that, following the terminology introduced by Cachazo and Skinner [3], Cachazo *et al.* [4] and Adamo and Mason [5], the graviton leaf amplitudes can be expressed in terms of multi-graviton wavefunctions defined on a supersymmetric extension of minitwistor space, **MT**, associated with Euclidean $AdS_3$. This reformulation facilitates the construction of a new class of graviton vertex operators, constructed exclusively from entities arising as worldsheet conformal primaries and Wess-Zumino-Novikov-Witten (WZNW) currents on the boundary of Euclidean $AdS_3$.

This advancement establishes a refined and symmetric correspondence between celestial CFT and $AdS_3$ string theory, wherein gluon and graviton vertex operators are treated on an equal footing. Their distinction is thereby confined to the number of supersymmetry generators and their respective gauge groups, simplifying and unifying the formalism.

The second line of inquiry is devoted to achieving a dynamical realisation of the framework initially proposed by de Boer and Solodukhin [6] and subsequently elaborated by Casali, Melton, and Strominger [7]. Specifically, we propose a generalisation of the WZNW model to a supersymmetric extension of Euclidean $AdS_3$, which holographically reproduces the tree-level MHV scattering amplitudes for gravitons in $\mathcal{N} = 8$ Supergravity and for gluons in $\mathcal{N} = 4$ supersymmetric Yang-Mills (SYM) theory.

The derivation of this holographic $AdS_3$ theory begins with the observation that the generating functional for tree-level MHV celestial amplitudes of gravitons and gluons can be constructed from

the chiral determinant representation of the WZNW action functional. To advance this formalism, we turn to the framework established by Abe, Nair, and Park [8] and Abe [9]. These authors demonstrated that the superspace constraints characterising the anti-self-dual sector of $\mathcal{N} = 8$ Supergravity and $\mathcal{N} = 4$ SYM theory can be embedded into a supersymmetric extension of twistor space, $\mathbf{PT}$. Utilising ideas from the harmonic superspace formalism developed by Galperin *et al.* [10], it was shown that the superspace constraints in this setting admit a so-called *chiral semi-analytic gauge*, which simplifies the constraint equations to those of a WZNW-like field theory.

Drawing inspiration from these ideas, and employing our observation that gluon and graviton leaf amplitudes naturally admit an interpretation on minitwistor space $\mathbf{MT}$, we proceed as follows. We solve the superspace constraints in the chiral semi-analytic gauge using the conformal primary basis, and, following Bu and Seet [11], we perform a scaling reduction from twistor space $\mathbf{PT}$ to minitwistor space $\mathbf{MT}$. This reduction yields explicit forms for the superpotentials which, when substituted into the WZNW-like action proposed by Abe, Nair, and Park [8], result in the generating functionals for graviton and gluon leaf amplitudes.

From this framework, we construct the effective action integral on $AdS_3$ for a WZNW-like field theory, and demonstrate that its associated Euler-Lagrange equations reproduce the superspace constraints in the chiral semi-analytic gauge. Moreover, we establish that the on-shell effective action precisely recovers the generating functionals for the leaf amplitudes.

*Notation.* We adopt the conventions of Weinberg [12] concerning the embedding space formalism, wherein $X^\mu$ denotes the Cartesian coordinates in $\mathbf{R}^4$, and $\hat{X}^\mu$ represents the restriction of $X^\mu$ to the standard hyperboloid $H_3^+ \subset \mathbf{R}^4$. The coordinates on the $(4|\mathcal{N})$-dimensional superspace $\mathbf{R}^{4|\mathcal{N}}$ are denoted by $\mathbb{X}^I := (X^\mu, \theta^{A\alpha}, \bar{\theta}_\alpha^{\dot{A}})$, where $\theta^{A\alpha}$ and $\bar{\theta}_\alpha^{\dot{A}}$ are Grassmann-valued, two-component spinors, and $\alpha = 1, ..., \mathcal{N}$. Our work here proceeds within the framework of DeWitt supermanifolds (cf., e.g., DeWitt [13], Rogers [14]; for analysis with super-numbers, see Berezin [15]. For an alternative to DeWitt supermanifolds, refer to Leites [16], Batchelor [17, 18]).

We are concerned, in what follows, with scattering processes involving gravitons in $\mathcal{N} = 8$ Supergravity and gluons in $\mathcal{N} = 4$ SYM theory. Lower-case Roman letters $i, j, k...$ will index the $n$ bosons involved in a scattering process. Let $z_i, \bar{z}_i \in \mathbf{CP}^1 \simeq S^2$ denote complex coordinates on the celestial sphere, which can also be parametrised by pairs of two-component spinors $\pi_i^A := (z_i, 1)^T$ and $\bar{\pi}_i^{\dot{A}} := (\bar{z}_i, 1)^T$, defining the standard null four-vector $q^\mu(z_i, \bar{z}_i) := (\sigma^\mu)_{A\dot{A}} \pi_i^A \bar{\pi}_i^{\dot{A}}$. The four-momentum of the $i$-th particle with frequency $s_i$ is parametrised by points $z_i, \bar{z}_i \in \mathbf{CP}^1$ on the

celestial sphere, using the standard null four-vector as:

$$p^\mu = s_i q^\mu(z_i, \bar{z}_i) = s_i \left(1 + z_i \bar{z}_i, z_i + \bar{z}_i, i(\bar{z}_i - z_i), 1 - z_i \bar{z}_i\right). \tag{1}$$

In writing the Berends-Giele-Kuijf (BGK) and Parke-Taylor formulae for tree-level MHV scattering amplitudes, it is useful to define the frequency-dependent pairs of two-component spinors $\nu_i^A := \sqrt{s_i} \pi_i^A$ and $\bar{\nu}_i^{\dot{A}} := \sqrt{s_i} \bar{\pi}_i^{\dot{A}}$, such that the four-momentum of the $i$-th graviton takes the form $p_i^{A\dot{A}} = \nu_i^A \bar{\nu}_i^{\dot{A}}$. In celestial CFT, the conformal weight attributed to the $i$-th graviton is denoted by $\Delta_i$.

## II.  $\mathcal{N} = 8$ SUPERGRAVITY

We begin in Subsection II A by revisiting the construction of tree-level MHV celestial amplitudes for gravitons as developed in Mol [19]. Subsequently, in Subsection II B, we refine the holographic dictionary proposed in Ogawa *et al.* [1] and Mol [2], which establishes a correspondence between (Euclidean) $AdS_3$ string theory and celestial CFT. Finally, in Subsection II C, we derive an expression for the celestial amplitude for gravitons in $\mathcal{N} = 8$ Supergravity, formulated in terms of multi-graviton wavefunctions on minitwistor space; this expression serves as our motivation for constructing a generating functional for graviton amplitudes from the chiral Dirac determinant.

### A.  Review

Our starting point is the Berends-Giele-Kuijf (BGK) formula for the tree-level scattering amplitude of $n$-gravitons in the maximally-helicity-violating configuration $1^{--}, 2^{--}, 3^{++}, ..., n^{++}$ (for $n \geq 5$) originally introduced by Berends, Giele, and Kuijf [20],

$$\mathcal{M}_n = \left(\frac{\kappa}{2}\right)^{n-2} \delta^{(4)} \left(\sum_{i=1}^{n} p_i^\mu\right) BGK_n + \mathscr{P}_{2,...,n-2}, \tag{2}$$

where:

$$BGK_n = \frac{\langle \nu_1, \nu_2 \rangle^8}{\langle \nu_1, \nu_2 \rangle ... \langle \nu_n, \nu_1 \rangle} \frac{1}{\langle \nu_n, \nu_1 \rangle \langle \nu_1, \nu_{n-1} \rangle \langle \nu_{n-1}, \nu_n \rangle} \prod_{k=2}^{n-2} \frac{[p_k | p_{k+1} + ... + p_{n-1} | p_n\rangle}{\langle \nu_k, \nu_n \rangle}. \tag{3}$$

Here, $\mathscr{P}_{2,...,n-2}$ denotes the permutation operator acting on the indices within the set $\{2, ..., n-2\}$.

*Fermionic Doublet on* $\mathbf{CP}^1$. Following the formalism introduced by Nair [21, 22], we incorporate an auxiliary fermionic doublet $(\hat{\chi}, \hat{\chi}^\dagger)$ defined on $\mathbf{CP}^1$ (identified as the celestial sphere). These are introduced through the mode expansions:

$$\hat{\chi}(z_i) := \sum_{k=0}^{\infty} b_k z_i^{-1-k}, \quad \hat{\chi}^\dagger(z_i) := \sum_{k=0}^{\infty} b_k^\dagger z_i^k, \tag{4}$$

where the annihilation and creation operators, $b_k$ and $b_k^\dagger$, respectively, satisfy the anti-commutation relations $\{b_k, b_{k'}^\dagger\} = \delta_{kk'}$ for $k, k' \geq 0$, and act on the vacuum state $|0\rangle$ as $b_k|0\rangle = 0$. Consequently, the two-point function of the doublet $(\hat{\chi}, \hat{\chi}^\dagger)$ is given by:

$$\langle 0|\hat{\chi}(z_i)\hat{\chi}^\dagger(z_j)|0\rangle = \frac{1}{z_i - z_j}. \tag{5}$$

In the subsequent discussion, we introduce the auxiliary two-component spinor $\lambda^A := (\lambda, 1)^T$. Integration will be performed over the complex variable $\lambda \in \mathbf{C}$, restricted to a small contour $C_n$ encircling the insertion point $z_n$ of the $n$-th graviton on the celestial sphere $S^2 \simeq \mathbf{CP}^1$.

We also define an analogous fermionic doublet $(\chi, \chi^\dagger)$ on the space of two-component spinors. Let $\omega_i^A := (\xi_i, \zeta_i)^T$ denote a two-component spinor, and consider an open neighbourhood $\mathcal{U}$ where $\zeta_i \neq 0$. In this setting, $\omega_i^A$ can be locally parametrised on $\mathbf{CP}^1$ by $z_i = \xi_i/\zeta_i$. We then define:

$$\chi(\omega_i) := \frac{1}{\zeta_i}\hat{\chi}(z_i), \quad \chi^\dagger(\omega_i) := \frac{1}{\zeta_i}\hat{\chi}^\dagger(z_i). \tag{6}$$

This definition ensures that the two-point function of the fermionic doublet $(\chi, \chi^\dagger)$ is given by:

$$\langle 0|\chi(\omega_i)\chi^\dagger(\omega_j)|0\rangle = \frac{1}{\langle \omega_i, \omega_j\rangle}. \tag{7}$$

*Nair's Vertex Operators.* Following the framework elaborated by Nair [22], we proceed to introduce the following operators:

$$\mathcal{Q}_i := e^{ip_i \cdot X}\chi^\dagger(\nu_i)\chi(\nu_i), \quad \mathcal{P}_i := \frac{1}{i}\frac{\bar{\nu}_i^{\dot{A}}\lambda^A}{\langle \nu_i, \lambda\rangle}\frac{\partial}{\partial X^{A\dot{A}}}e^{ip_i \cdot X}, \tag{8}$$

where $\lambda^A := (\lambda, 1)^T$ represents the auxiliary two-component spinor.

As demonstrated in detail in Mol [19], these operators satisfy the identity:

$$\int \frac{d^4X}{(2\pi)^4}\oint_{C_n}\frac{d\lambda}{2\pi i}\langle\lambda|\mathcal{Q}_1\left(\prod_{k=2}^{n-2}\mathcal{P}_k\right)\mathcal{Q}_{n-1}\mathcal{Q}_n|\lambda\rangle \tag{9}$$

$$= -\delta^{(4)}\left(\sum_{i=1}^{n}p_i^\mu\right)\frac{1}{\langle\nu_n,\nu_1\rangle\langle\nu_1,\nu_{n-1}\rangle\langle\nu_{n-1},\nu_n\rangle}\prod_{k=2}^{n-2}\frac{[p_k|p_{k+1}+...+p_{n-1}|p_n\rangle}{\langle\nu_k,\nu_n\rangle}. \tag{10}$$

Consequently, the $n$-graviton amplitude can be expressed as:

$$\mathcal{M}_n = -\left(\frac{\kappa}{2}\right)^{n-2}\frac{\langle\nu_1,\nu_2\rangle^8}{\langle\nu_1,\nu_2\rangle...\langle\nu_n,\nu_1\rangle}\int\frac{d^4X}{(2\pi)^4}\oint_{C_n}\frac{d\lambda}{2\pi i}\langle\lambda|\mathcal{Q}_1\left(\prod_{k=2}^{n-2}\mathcal{P}_k\right)\mathcal{Q}_{n-1}\mathcal{Q}_n|\lambda\rangle \tag{11}$$

$$+ \mathscr{P}_{2,...,n-2}. \tag{12}$$

*Celestial Amplitudes.* The present formalism now departs from the construction proposed by Nair [22], as we now prepare to perform the Mellin transform and derive the celestial graviton amplitude. Reformulating Eq. (11) in terms of the frequencies $s_i$ and two-component spinors $\pi_i^A := (z_i, 1)^T$, the amplitude takes the form:

$$\mathcal{M}_n = -\left(\frac{\kappa}{2}\right)^{n-2} \prod_{i=1}^{n} s_i^{e_i} \frac{\langle \pi_1, \pi_2 \rangle^8}{\langle \pi_1, \pi_2 \rangle \dots \langle \pi_n, \pi_1 \rangle} \int \frac{d^4 X}{(2\pi)^4} \oint_{C_n} \frac{d\lambda}{2\pi i} \langle \lambda | \mathcal{Q}_1 \left( \prod_{k=2}^{n-2} \mathcal{P}_k \right) \mathcal{Q}_{n-1} \mathcal{Q}_n | \lambda \rangle \quad (13)$$

$$+ \mathscr{P}_{2,\dots,n-2}, \quad (14)$$

where the exponents for the chosen MHV configuration are given by $e_1 = e_2 = 3$ and $e_3 = \dots = e_n = -1$. Noting that:

$$\mathcal{Q}_i = \frac{1}{s_i} e^{i s_i q(z_i, \bar{z}_i) \cdot X} \hat{\chi}^\dagger (z_i) \hat{\chi}(z_i), \quad \mathcal{P}_i = \frac{1}{i} \frac{\bar{\pi}_i^{\dot{A}} \lambda^A}{\langle \pi_i, \lambda \rangle} \frac{\partial}{\partial X^{A\dot{A}}} e^{i s_i q(z_i, \bar{z}_i) \cdot X}. \quad (15)$$

the Mellin-transformed operators are given by:

$$\widehat{\mathcal{Q}}_i := \int_{(0,\infty)} \frac{ds_i}{s_i} s_i^{\Delta_i} \left( s_i^{e_i} \mathcal{Q}_i \right) = \phi_{2h_i} \left( z_i, \bar{z}_i | X \right) \hat{\chi}^\dagger (z_i) \hat{\chi}(z_i), \quad (16)$$

$$\widehat{\mathcal{P}}_i := \int_{(0,\infty)} \frac{ds_i}{s_i} s_i^{\Delta_i} \left( s_i \mathcal{P}_i \right) = \frac{1}{i} \frac{\bar{\pi}_i^{\dot{A}} \lambda^A}{\langle \pi_i, \lambda \rangle} \frac{\partial}{\partial X^{A\dot{A}}} \phi_{2h_i} \left( z_i, \bar{z}_i | X \right). \quad (17)$$

Here, following Pasterski and Shao [23], we introduced the celestial conformal primary wavefunction for massless scalars:

$$\phi_{2h_i} \left( z_i, \bar{z}_i | X \right) := \frac{\Gamma \left( 2h_i \right)}{\left( \varepsilon - iq \left( z_i, \bar{z}_i \right) \cdot X \right)}, \quad (18)$$

with $\varepsilon > 0$ serving as a regulator. For the MHV configuration $1^{--}, 2^{--}, 3^{++}, \dots, n^{++}$, the scaling dimensions are defined as follows:

$$2h_i := \Delta_i + e_i - 1, \ \ i \in \{1, n-1, n\}; \ \ 2h_i := \Delta_i + e_i, \ \ i \in \{2, \dots, n-2\}. \quad (19)$$

The celestial $n$-amplitude is the $\varepsilon$-regulated Mellin transform of $\mathcal{M}_n$:

$$\widehat{\mathcal{M}}_n = \prod_{k=1}^{n} \int_{(0,\infty)} \frac{ds_k}{s_k} s_k^{\Delta_k} e^{-\varepsilon s_k} \mathcal{M}_n. \quad (20)$$

Therefore:

$$\widehat{\mathcal{M}}_n = -\left(\frac{\kappa}{2}\right)^{n-2} \frac{\langle \pi_1, \pi_2 \rangle^8}{\langle \pi_1, \pi_2 \rangle \dots \langle \pi_n, \pi_1 \rangle} \int \frac{d^4 X}{(2\pi)^4} \oint_{C_n} \frac{d\lambda}{2\pi i} \langle \lambda | \widehat{\mathcal{Q}}_1 \left( \prod_{k=2}^{n-2} \widehat{\mathcal{P}}_k \right) \widehat{\mathcal{Q}}_{n-1} \widehat{\mathcal{Q}}_n | \lambda \rangle \quad (21)$$

$$+ \mathscr{P}_{2,\dots,n-2}. \quad (22)$$

*Supersymmetry.* The transition from General Relativity to $\mathcal{N} = 8$ Supergravity begins with the following observation. Let $\theta^{A\alpha}$ ($\alpha = 1, ..., 8$) denote Grassmann-valued two-component spinors normalised such that:

$$\int d^2\theta \theta_A^\alpha \theta_B^\alpha = \varepsilon_{AB}, \tag{23}$$

where the Berezin integral is defined as per Berezin [15] and DeWitt [13]. By defining the coefficients:

$$\xi_i^\alpha := \theta^{A\alpha} \pi_{iA} \quad (1 \leq i \leq n), \tag{24}$$

one obtains:

$$\int d^{16}\theta \prod_{\alpha=1}^{8} \xi_i^\alpha \xi_j^\alpha = \langle \pi_i, \pi_j \rangle^8. \tag{25}$$

Then, Eq. (21) can be rewritten as:

$$\widehat{\mathcal{M}}_n = -\left(\frac{\kappa}{2}\right)^{n-2} \int d^{16}\theta \left(\prod_{\alpha=1}^{8} \xi_1^\alpha \xi_2^\alpha\right) \prod_{i=1}^{n} \frac{1}{\langle \pi_i, \pi_{i+1} \rangle} \tag{26}$$

$$\int \frac{d^4 X}{(2\pi)^4} \oint_{C_n} \frac{d\lambda}{2\pi i} \langle \lambda | \widehat{\mathcal{Q}}_1 \left(\prod_{k=2}^{n-2} \widehat{\mathcal{P}}_k\right) \widehat{\mathcal{Q}}_{n-1} \widehat{\mathcal{Q}}_n | \lambda \rangle + \mathscr{P}_{2,...,n-2}. \tag{27}$$

Here, we set $\pi_{n+1} := \pi_1$, so that $\prod_{k=1}^{n} \langle \pi_k, \pi_{k+1} \rangle = z_{12} z_{23} ... z_{n1}$.

This naturally motivates the introduction of the $\mathcal{N} = 8$ superfield:

$$\Psi_i := h^-(z_i, \bar{z}_i) + \xi^\alpha \tilde{\lambda}_\alpha + \frac{1}{2} \xi^\alpha \xi^\beta \widetilde{\varphi}_{\alpha\beta} + ... + h^+(z_i, \bar{z}_i) \prod_{\alpha=1}^{8} \xi_i^\alpha, \tag{28}$$

which contains the degrees of freedom $(h^\pm, \tilde{\lambda}_\alpha, \widetilde{\phi}_{\alpha\beta}, ...)$ of $\mathcal{N} = 8$ Supergravity. Here, $h_i^- := h^-(z_i, \bar{z}_i)$ and $h_i^+ := h^+(z_i, \bar{z}_i)$ correspond to the vacuum expectation values of the annihilation operators for gravitons of negative and positive helicities, respectively. The dependence of $\Psi_i$ on the holomorphic and anti-holomorphic coordinates $z_i, \bar{z}_i$ is encoded through the coefficients $\xi_i^\alpha$ ($\alpha = 1, ..., 8$) defined in Eq. (24).

Therefore, we pass from the celestial amplitude $\widehat{\mathcal{M}}_n$ to the *celestial partial super-amplitude* $\widehat{\mathbb{M}}_n$:

$$\widehat{\mathbb{M}}_n = -\frac{1}{(2\pi)^4} \left(\frac{\kappa}{2}\right)^{n-2} \int d^4 X \int d^{16}\theta \oint_{C_n} \frac{d\lambda}{2\pi i} \langle \lambda | \widehat{\mathcal{Q}}_1 \left(\prod_{k=2}^{n-2} \widehat{\mathcal{P}}_k\right) \widehat{\mathcal{Q}}_{n-1} \widehat{\mathcal{Q}}_n | \lambda \rangle \prod_{\ell=1}^{n} \frac{\Psi_\ell}{\langle \pi_\ell, \pi_{\ell+1} \rangle} \tag{29}$$

$$+ \mathscr{P}_{2,...,n-2}. \tag{30}$$

*Factorisation of Celestial Wavefunctions.* To incorporate the formalism of leaf amplitudes as developed by Casali, Melton, and Strominger [7] and Melton, Sharma, and Strominger [24], we first analytically continue $\widehat{\mathbb{M}}_n$ from Lorentzian to Kleinian signature via the transformation $X^2 \mapsto -iX^2$. Then, the celestial amplitude must be expressed as a weighted integral over the leaves of the natural hyperbolic foliation of Klein space $\mathbf{R}^{(2,2)}$, in terms of the Kleinian hyperboloid $\mathbf{H}_3 := AdS_3/\mathbf{Z}$. Achieving this requires factoring the celestial wavefunctions $\phi_{2h_i}(z_i, \bar{z}_i | X)$ contained within the operators $\widehat{\mathcal{Q}}_i$ and $\widehat{\mathcal{P}}_i$ outside the correlator $\langle \lambda | ... | \lambda \rangle$.

To facilitate this, we define the following pair of operators:

$$\mathsf{A}_i := \exp\left(q(z_i, \bar{z}_i) \cdot y\, \mathsf{P}_i\right) \hat{\chi}^\dagger(z_i) \hat{\chi}(z_i), \quad \mathsf{B}_i := \frac{\bar{\pi}_i^{\dot{A}} \lambda^A}{\langle \pi_i, \lambda \rangle} \frac{\partial}{\partial y^{A\dot{A}}} \exp\left(q(z_i, \bar{z}_i) \cdot y\, \mathsf{P}_i\right), \tag{31}$$

where $\mathsf{P}_i$ is a weight-shifting operator that acts on celestial wavefunctions by:

$$\mathsf{P}_i \phi_{2h_j}(z_j, \bar{z}_j | X) := \phi_{2h_j + \delta_{ij}}(z_j, \bar{z}_j | X), \tag{32}$$

and $y^{A\dot{A}}$ is a four-vector that parametrises the family of operators $\mathsf{A}_i, \mathsf{B}_i$.

Consider now the following identity:

$$\frac{1}{i} \frac{\bar{\pi}_i^{\dot{A}} \lambda^A}{\langle \pi_i, \lambda \rangle} \frac{\partial}{\partial X^{A\dot{A}}} \phi_{2h_j}(z_j, \bar{z}_j | X) \tag{33}$$

$$= \int d^4 y\, \delta^{(4)}(y) \frac{\bar{\pi}_i^{\dot{A}} \lambda^A}{\langle \pi_i, \lambda \rangle} \frac{\partial}{\partial y^{A\dot{A}}} \exp\left(q(z_j, \bar{z}_j) \cdot y\, \mathsf{P}_j\right) \phi_{2h_j}(z_j, \bar{z}_j | X). \tag{34}$$

For the sake of notational economy, henceforth we shall omit the integral $\int d^4 y\, \delta^{(4)}(y)$. Accordingly, all subsequent equalities involving the operators $\mathsf{A}_i$ and $\mathsf{B}_i$ should be understood as being evaluated at $y^{A\dot{A}} = 0$.

Under this convention, the celestial super-amplitude can be reformulated in terms of these operators as follows:

$$\widehat{\mathbb{M}}_n = -\frac{1}{(2\pi)^4} \left(\frac{\kappa}{2}\right)^{n-2} \oint_{C_n} \frac{d\lambda}{2\pi i} \langle \lambda | \mathsf{A}_1 \left(\prod_{k=2}^{n-2} \mathsf{B}_k\right) \mathsf{A}_{n-1} \mathsf{A}_n | \lambda \rangle \tag{35}$$

$$\int d^4 X \int d^{16}\theta \prod_{\ell=1}^{n} \frac{\phi_{2h_\ell}(z_\ell, \bar{z}_\ell | X) \Psi_\ell}{\langle \pi_\ell, \pi_{\ell+1} \rangle} + \mathscr{P}_{2,\ldots,n-2}. \tag{36}$$

As desired, *this representation explicitly factors out the product $\prod_\ell \phi_{2h_\ell}(z_\ell, \bar{z}_\ell | x)$ of celestial wavefunctions from the correlator $\langle \lambda | \cdots | \lambda \rangle$.*

## B. Euclidean $AdS_3$ String Theory and Graviton Leaf Amplitudes

We now proceed to derive a novel expression for the tree-level MHV celestial super-amplitude for gravitons within $\mathcal{N} = 8$ Supergravity. This formula will facilitate the construction of an improved

correspondence between (Euclidean) $AdS_3$ string theory and celestial CFT, as originally suggested by Ogawa *et al.* [1] and Mol [2].

*Celestial Leaf Amplitudes.* Utilising the formalism of celestial leaf amplitudes as developed by Melton, Sharma, and Strominger [24], Melton *et al.* [25] and reviewed in the Appendix of Mol [19], we examine the spacetime integral, $\int d^4 X \, (\cdot)$, appearing in Eq. (35). Under the simplifying assumption that all gravitons are outgoing, this integral admits the following representation:

$$\int d^4 X \prod_{k=1}^{n} \phi_{2h_k}\left(z_k, \bar{z}_k \middle| X\right) = 2\pi\delta\left(\beta\right) \int_{\mathbf{H}_3} d^3 x \prod_{k=1}^{n} G_{2h_k}(z_k, \bar{z}_k | x) + (\bar{z}_k \to -\bar{z}_k) \tag{37}$$

where the function $x \in \mathbf{H}_3 \mapsto G_\Delta\left(z, \bar{z} \middle| x\right)$ is given by:

$$G_\Delta\left(z, \bar{z} \middle| x\right) := \frac{\Gamma\left(\Delta\right)}{\left(\varepsilon - iq\left(z, \bar{z}\right) \cdot x\right)^\Delta}, \tag{38}$$

and corresponds to the bulk-to-boundary Green's function for the covariant Laplacian on the hyperboloid $\mathbf{H}_3$ (cf. Teschner [26, Appendix A], Penedones [27] and Costa, Gonçalves, and Penedones [28]). The symbol $(\bar{z}_k \to -\bar{z}_k)$ indicates the repetition of the first term under the replacement $\bar{z}_k \mapsto -\bar{z}_k$ for all $1 \le k \le n$. The parameter $\beta$, defined by:

$$\beta := 2\sum_{i=1}^{n} h_i - 4, \tag{39}$$

encodes the total scaling dimension associated with the scattering process under consideration. Accordingly, the integral $\int d^3 x \, (\cdot)$ over the hyperboloid $\mathbf{H}_3$ appearing on the right-hand side of Eq. (37) can be interpreted physically as a contact Feynman-Witten diagram representing the propagation of massless scalar particles in Euclidean $AdS_3$.

*The Weight-shifting Polynomial Operator.* To simplify subsequent computations, we define the following operator, acting on the set $\{\phi_{2h_i}(z_i, \bar{z}_i | X)\}$ of celestial conformal primaries:

$$\oint dh_1...dh_n := \frac{i}{2\pi} \oint_{C_n} d\lambda \langle\lambda|\mathsf{A}_1 \left(\prod_{k=2}^{n-2} \mathsf{B}_k\right) \mathsf{A}_{n-1}\mathsf{A}_n|\lambda\rangle. \tag{40}$$

Recalling the definitions of the operators $\mathsf{A}_i$ and $\mathsf{B}_i$, as introduced in Eq. (31), and employing an inductive argument, *the operator $\oint dh_1...dh_n$ can be expanded as a polynomial in the weight-shifting operators $\mathsf{P}_i$*:

$$\oint dh_1...dh_n = \frac{1}{(z_n - z_1)(z_1 - z_{n-1})(z_{n-1} - z_n)} \left(\prod_{i=2}^{n-2} \sum_{j_i=i+1}^{n}\right) \prod_{k=2}^{n-2} \frac{(\bar{z}_k - \bar{z}_{j_k})(z_{j_k} - z_n)}{z_k - z_n} \mathsf{P}_{j_k}. \tag{41}$$

Since $\mathsf{P}_1, ..., \mathsf{P}_n$ acts by shifting the scaling dimensions $2h_1, ..., 2h_n$, this expression provides a justification for the mnemonic symbol $\oint dh_1...dh_n$ employed in its definition.

Finally, the celestial super-amplitude $\widehat{\mathbb{M}}_n$ can be elegantly rewritten as:

$$\widehat{\mathbb{M}}_n = \frac{1}{(2\pi)^3} \left(\frac{\kappa}{2}\right)^{n-2} \oint dh_1 ... dh_n \delta\left(\beta\right) M_n + \left(\bar{z}_k \rightarrow -\bar{z}_k\right) + \mathscr{P}_{2,...,n-2} \tag{42}$$

where, following Melton, Sharma, and Strominger [24], the leaf amplitude $M_n$ is defined by:

$$M_n := \int_{\mathbf{H}_3} d^3x \int d^{16}\theta \prod_{k=1}^{n} G_{2h_k}\left(z_k, \bar{z}_k \big| x\right) \frac{\Psi_k}{\langle \pi_k, \pi_{k+1} \rangle}. \tag{43}$$

*Remark.* The foregoing expression describes the tree-level MHV leaf amplitude for gravitons within $\mathcal{N} = 8$ Supergravity, and notably, it relies exclusively on the kinematical data parametrised by points $x \in \mathbf{H}_3$ on the hyperboloid, and $(z_i, \bar{z}_i) \in \mathbf{CP}^1$, representing coordinates on the celestial sphere. This construction thus achieves the sought-after reformulation of the celestial super-amplitude in the desired geometric and kinematic variables.

*Remark.* The graviton leaf amplitude $M_n$ can be equivalently expressed in the form:

$$M_n = \prod_{i=1}^{n} \int d^2\mu_i \frac{\Gamma\left(2h_i\right)}{(-i\mu_i \cdot \bar{\pi}_i)} \int_{\mathbf{H}_3} d^3x \int d^{16}\theta \, \psi, \tag{44}$$

where:

$$\psi := \prod_{i=1}^{n} \delta^{(2)}\left(\mu_{\dot{A}i} - \pi_i^A x_{A\dot{A}}\right) \frac{\Psi_i}{\langle \pi_i, \pi_{i+1} \rangle}. \tag{45}$$

Following the terminology introduced by Cachazo and Skinner [3], Cachazo *et al.* [4] and Adamo and Mason [5], the function $\psi$ may be understood as the *multi-graviton wavefunction* constructed on the minitwistor space $(\mu_{\dot{A}i}, \pi_i^A) \in \mathbf{MT}$ associated with the hyperboloid $\mathbf{H}_3$. From this perspective, the geometry of minitwistor space provides a framework for encoding the kinematic data of *leaf* amplitudes. For a review of the minitwistor space $\mathbf{MT}$ in the context of celestial holography, cf. Bu and Seet [11].

*Euclidean AdS$_3$ String Theory.* The connection between celestial CFT and (Euclidean) *AdS$_3$* string theory can be introduced through the following observations. Let $\Phi^{h_i}\left(w_i \big| z_i\right)$ denote the conformal primary fields of the level-$k$ $H_3^+$-WZNW model, characterised by spin $h_i$, where $w_i, \bar{w}_i \in \Sigma$ are worldsheet coordinates, and $z_i, \bar{z}_i \in \mathbf{CP}^1$ are coordinates on the celestial sphere. For notational convenience, we suppress the dependence of the conformal primaries on the anti-holomorphic coordinates $\bar{w}_i$ and $\bar{z}_i$.

It was demonstrated by Teschner [26] and subsequently refined by Ribault and Teschner [29] that, in the minisuperspace limit, defined by $k \rightarrow \infty$, the correlation functions of the conformal

primaries in the $H_3^+$-WZNW model can be expressed in terms of the bulk-to-boundary Green's functions on $\mathbf{H}_3$ as follows:

$$\lim_{k \to \infty} \prod_{i=1}^{n} \int d^2 w_i \Gamma \left( 2h_i \right) \left\langle \Phi^{h_1} \left( w_1 | z_1 \right) ... \Phi^{h_n} \left( w_n | z_n \right) \right\rangle = \int_{\mathbf{H}_3} d^3 x \prod_{i=1}^{n} G_{2h_i} \left( z_i, \bar{z}_i | x \right). \qquad (46)$$

To elaborate: the minisuperspace limit of the correlation functions of the primary fields in the $H_3^+$-WZNW model yields the computation of a contact Feynman-Witten diagram on the three-dimensional hyperboloid.

We now consider (Euclidean) $AdS_3$ string theory formulated on the target space $AdS_3 \times X$, where $X$ is a compact manifold. Let $J^a \left( z \right)$ be a level-one $SO \left( 2N \right)$ WZNW current algebra defined on the boundary of $AdS_3$. It is postulated that $J^a \left( z \right)$ originates from the CFT on $X$, henceforth referred to as the $X$-CFT.

Let $\mathsf{T}^a$ represent the generators of the adjoint representation of $SO \left( 2N \right)$, normalised such that $\mathrm{Tr} \left( \mathsf{T}^a \mathsf{T}^b \right) = 2\delta^{ab}$, with the commutation relations $\left[ \mathsf{T}^a, \mathsf{T}^b \right] = i f^{abc} \mathsf{T}^c$. At leading-trace order, the correlators of the WZNW currents $J^a \left( z \right)$ satisfy the following identity:

$$\left\langle J^{a_1} (z_1) ... J^{a_n} \left( z_n \right) \right\rangle \sim \frac{\mathrm{Tr} \left( \mathsf{T}^{a_1} ... \mathsf{T}^{a_n} \right)}{z_{12} z_{23} ... z_{n1}} = \mathrm{Tr} \prod_{i=1}^{n} \frac{\mathsf{T}^{a_i}}{\langle \pi_i, \pi_{i+1} \rangle}. \qquad (47)$$

Hereafter, all equations involving the correlation functions of the WZNW currents $J^a \left( z \right)$ are to be understood at the leading-trace order.

Subsequently, we define the currents $\mathcal{J} \left( z \right) \coloneqq \mathsf{T}^a J^a \left( z \right)$, such that:

$$\left\langle \mathcal{J} \left( z_1 \right) ... \mathcal{J} \left( z_n \right) \right\rangle = \mathcal{N} \prod_{i=1}^{n} \frac{1}{\langle \pi_i, \pi_{i+1} \rangle}. \qquad (48)$$

The proportionality factor is given by $\mathcal{N} \coloneqq \mathrm{Tr} \left( \mathsf{T}^{a_1} ... \mathsf{T}^{a_n} \right)^2$. Building on the foundational works of Giveon, Kutasov, and Seiberg [30], Dolan and Witten [31], Dolan [32], let $W_X$ denote a spinless operator in the $X$-CFT. One can construct physical vertex operators of the form $W_X V_{jm\bar{m}}$, where $V_{jm\bar{m}}$ represents a worldsheet vertex operator.

In particular, we define the *celestial vertex operators for gravitons* as:

$$\mathcal{V}_i \coloneqq \int d^2 w_i \Gamma \left( 2h_i \right) \mathcal{J} \left( z_i \right) \Psi_i \Phi^{h_i} \left( w_i | z_i \right). \qquad (49)$$

Finally, the *leaf amplitude $M_n$* for graviton scattering in $\mathcal{N} = 8$ Supergravity can then be expressed as the minisuperspace limit ($k \to \infty$) of (Euclidean) $AdS_3$ string theory as:

$$M_n = \frac{1}{\mathcal{N}} \lim_{k \to \infty} \int d^{16}\theta \left\langle \mathcal{V}_1 ... \mathcal{V}_n \right\rangle. \qquad (50)$$

This result provides a refined formulation of the correspondence between the sector of celestial CFT encoding the tree-level MHV graviton scattering amplitudes and (Euclidean) $AdS_3$ string theory. Notably, in contrast to our earlier proposal (Mol [2]), the celestial vertex operators $\mathcal{V}_i$ for gravitons are entirely constructed from objects derived as solutions to the string theory equations of motion, either worldsheet conformal primaries or currents from the $X$-CFT. Crucially, it is no longer necessary to "dress" the operators $\mathcal{V}_i$ with any additional *ad hoc* components.

## C. Generating Functional

We now undertake the derivation of a generating functional for the graviton leaf amplitude $M_n$ using the WZNW action. The result herein constitutes a key step in our construction of a holographic $AdS_3$ field theory dual to the sector of celestial CFT containing $\mathcal{N} = 8$ Supergravity. The derivation rests upon two basic observations.

Firstly, let $\mathcal{S}[G]$ denote the action integral of the level-one $SO(2N)$ WZNW model defined on $\mathbf{CP}^1$. The dynamical variable $G = G(z, \bar{z})$ is a $\mathbf{CP}^1$-dependent matrix-valued field in the Lie group $SO(2N)$. From $G$, we construct a differential one-form $\boldsymbol{\omega}$ on $\mathbf{CP}^1$,

$$\boldsymbol{\omega} := \omega_z dz + \omega_{\bar{z}} d\bar{z}, \tag{51}$$

where the holomorphic and anti-holomorphic components are defined as follows:

$$\omega_z := -\left(\partial_z G\right) G^{-1}, \;\; \omega_{\bar{z}} := -\left(\partial_{\bar{z}} G\right) G^{-1}. \tag{52}$$

As reviewed by Nair [33], if $G$ satisfies the Euler-Lagrange equations associated with the WZNW action $\mathcal{S}[G]$, then the one-form $\boldsymbol{\omega}$ obeys the condition:

$$\mathcal{F}[\boldsymbol{\omega}] := \partial_z \omega_{\bar{z}} - \partial_{\bar{z}} \omega_z + [\omega_z, \omega_{\bar{z}}] = 0. \tag{53}$$

This equation suggests that $\boldsymbol{\omega}$ can be interpreted as the gauge potential corresponding to a connection one-form on $\mathbf{CP}^1$, valued in the Lie algebra $\mathfrak{so}(2N)$. The term $\mathcal{F}[\boldsymbol{\omega}]$ then represents the curvature two-form, which quantifies the field strength of the gauge potential $\boldsymbol{\omega}$. Consequently, the vanishing of the field strength, $\mathcal{F}[\boldsymbol{\omega}] = 0$, can be identified with the "flatness" condition for the connection defined by $\boldsymbol{\omega}$.

The second observation proceeds as follows. Let $\mathcal{D}_z := \partial_z + [\omega_z, \cdot]$ denote the holomorphic component of the gauge-covariant derivative operator induced by the connection $\boldsymbol{\omega}$. As explained by Nair [33], the WZNW action $\mathcal{S}[G]$ admits an alternative representation as a chiral determinant:

$$\mathcal{S}[G] = \mathcal{S}[\boldsymbol{\omega}] := \text{Tr} \log \mathcal{D}_z - \text{Tr} \log \omega_z. \tag{54}$$

This expression can be formally expanded as:

$$\mathcal{S}[\boldsymbol{\omega}] = \sum_{m \geq 2} \frac{(-1)^{m+1}}{m} \int \frac{d^2 z_1}{\pi} ... \int \frac{d^2 z_m}{\pi} \mathrm{Tr} \left( \frac{\omega_z \left( z_1, \bar{z}_1 \right) ... \omega_z \left( z_m, \bar{z}_m \right)}{z_{12} z_{23} ... z_{m1}} \right). \tag{55}$$

Now, consider the possibility of lifting the connection one-form $\boldsymbol{\omega}$ from $\mathbf{CP}^1$ to a holomorphic extension over the $\mathcal{N} = 8$ supersymmetric hyperboloid $\mathbf{H}_3$. Denote this lifted connection by $\boldsymbol{\Omega} := \Omega_z dz + \Omega_{\bar{z}} dz$, with the holomorphic component defined as:

$$\Omega_z(z_i, \bar{z}_i | x, \theta, \bar{\theta}) := \int_{\mathcal{P}} \frac{d\Delta_i}{2\pi i} G_{2h_i} \left( z_i, \bar{z}_i | x \right) \Psi_i. \tag{56}$$

Here, $\mathcal{P} := 1 + i\mathbf{R}$, and the scaling dimension $2h_i$ depends affinely on $\Delta_i$, as specified in Eq. (19). Substituting this expression into the expanded form of $\mathcal{S}[\boldsymbol{\omega}]$, we obtain:

$$\mathcal{S}[\boldsymbol{\Omega}] = \sum_{m \geq 2} \frac{(-1)^{m+1}}{m} \left( \prod_{k=1}^{n} \int_{\mathcal{P}} \frac{d\Delta_k}{2\pi i} \int \frac{d^2 z_k}{\pi} \right) \prod_{i=1}^{n} \frac{\Omega_z(z_i, \bar{z}_i | x, \theta, \bar{\theta})}{\langle \pi_i, \pi_{i+1} \rangle}. \tag{57}$$

We then define the "effective" action functional by:

$$\exp(\mathcal{W}[\boldsymbol{\Omega}]) := \int_{\mathbf{H}_3} d^3 x \int d^{16}\theta \, \mathcal{S}[\boldsymbol{\Omega}]. \tag{58}$$

Taking the functional derivatives of $e^{\mathcal{W}[\boldsymbol{\Omega}]}$ with respect to $h_i^{\pm}$, the vacuum expectation values of annihilation operators for gravitons with positive and negative helicities (contained in the $\mathcal{N} = 8$ superfield $\Psi_i$, as defined in Eq. (28), Subsection II A) yields:

$$M_n = \frac{\delta}{\delta h_1^-} \frac{\delta}{\delta h_2^-} \frac{\delta}{\delta h_3^+} ... \frac{\delta}{\delta h_n^+} \exp(\mathcal{W}[\boldsymbol{\Omega}]) \Big|_{\boldsymbol{\Omega}=0}. \tag{59}$$

Thus, the effective action $\mathcal{W}[\boldsymbol{\Omega}]$ serves as the generating functional for the graviton leaf amplitude $M_n$ in $\mathcal{N} = 8$ Supergravity.

Consequently, the construction of the holographic $AdS_3$ theory proceeds through the following steps:

1. *Postulate an Action Integral.* Introduce an action $\mathcal{I}[G, \boldsymbol{\Omega}]$ such that its variation with respect to the matrix field $G$ imposes the "flatness" condition $\mathcal{F}[\boldsymbol{\Omega}] = 0$, and its on-shell effective action coincides with $\mathcal{W}[\boldsymbol{\Omega}]$.

2. *Lift the Connection.* Establish a mathematical procedure to justify the lifting of the $\mathbf{CP}^1$ connection $\boldsymbol{\omega}$ to the holomorphically extended $\mathcal{N} = 8$ supersymmetric hyperboloid $\mathbf{H}_3$.

The procedure for lifting the $\mathbf{CP}^1$ connection $\boldsymbol{\omega}$ to its extension $\boldsymbol{\Omega}$ on $\mathbf{H}_3$ is predicated upon an important insight by Abe, Nair, and Park [8] and Abe [9]. Drawing inspiration from the harmonic

superspace formalism, comprehensively reviewed in Galperin *et al.* [10], these authors demonstrated that the superspace constraints of $\mathcal{N} = 4$ supersymmetric Yang-Mills theory and the anti-self-dual sector of $\mathcal{N} = 8$ Supergravity can be embedded into a supersymmetric generalisation of twistor space **PT**. By imposing the so-called *chiral semi-analytic gauge*, these constraints are shown to reduce exactly to the "flatness" condition $\mathcal{F}[\mathbf{\Omega}] = 0$ introduced in the preceding discussion. Building upon their work, we shall prove that a scaling-reduction of the solution to the *analyticity constraint* results precisely in the lifting $\mathbf{\Omega}$ of the $\mathbf{CP}^1$ connection $\boldsymbol{\omega}$ postulated in Eq. (56).

## III.   $\mathcal{N} = 4$ SUPERSYMMETRIC YANG-MILLS THEORY

### A.   Review

The starting point of our analysis is the Parke-Taylor (PT) formula, which provides an elegant representation for the tree-level scattering amplitude of $n$ gluons in a MHV configuration. Originally proposed by Parke and Taylor [34] and subsequently given a rigorous proof by Berends and Giele [35], this formula expresses the amplitude in the helicity configuration $1^-, 2^-, 3^+, ..., n^+$ as[1]:

$$\mathcal{A}_n^{a_1...a_n} = ig^{n-2}\delta^{(4)}\left(\sum_{i=1}^{n} p_i^\mu\right)(PT)_n^{a_1...a_n}, \tag{60}$$

where:

$$(PT)_n^{a_1...a_n} = \frac{\langle \nu_1, \nu_2 \rangle^4 \, \mathrm{Tr}\left(\mathsf{R}^{a_1}\mathsf{R}^{a_n}...\mathsf{R}^{a_n}\right)}{\langle \nu_1, \nu_2 \rangle \langle \nu_2, \nu_3 \rangle ... \langle \nu_n, \nu_1 \rangle}. \tag{61}$$

Here, $g$ denotes the Yang-Mills coupling constant, while $\delta^{(4)}$ imposes the conservation of four-momentum across the scattering process. We denote by $\mathsf{R}^a$ the generators of the adjoint representation of the gauge group $SO(\tilde{N})$, with their normalisation specified by $\mathrm{Tr}\left(\mathsf{R}^a\mathsf{R}^b\right) = 2\delta^{ab}$. These generators satisfy the commutation relations $[\mathsf{R}^a, \mathsf{R}^b] = i\mathsf{C}^{abc}\mathsf{R}^c$, where $\mathsf{C}^{abc}$ are the structure constants of $SO(\tilde{N})$. *For the sake of precision, we emphasise that the gauge group under consideration in this section differs from that employed in the preceding analysis.*

The scattering amplitude $\mathcal{A}_n^{a_1...a_n}$ can be reformulated in terms of the gluon frequencies $s_i$ and the two-component spinors $\pi_i^A := (z_i, 1)^T$ as follows:

$$\mathcal{A}_n^{a_1...a_n} = ig^{n-2}\int \frac{d^4X}{(2\pi)^4}\prod_{i=1}^{n} s_i^{e_i} e^{is_i q(z_i, \bar{z}_i)\cdot X}\frac{\langle \pi_1, \pi_2 \rangle^4 \, \mathrm{Tr}\left(\mathsf{R}^{a_1}\mathsf{R}^{a_n}...\mathsf{R}^{a_n}\right)}{\langle \pi_1, \pi_2 \rangle \langle \pi_2, \pi_3 \rangle ... \langle \pi_n, \pi_1 \rangle}. \tag{62}$$

In this representation, the exponents $e_1 = e_2 = 1$ and $e_3 = ... = e_n = -1$ correspond to the MHV configuration $1^-, 2^-, 3^+, ..., n^+$.

––––––––––

[1] For modern pedagogical introductions, see Elvang and Huang [36], Badger *et al.* [37].

*Supersymmetry.* The transition to $\mathcal{N} = 4$ SYM theory is facilitated by the introduction of Grassmann-valued two-component spinors $\theta^{A\alpha}$ ($\alpha = 1, ..., 4$), normalised as $\int d^2\theta \theta_A^\alpha \theta_B^\alpha = \varepsilon_{AB}$. From these spinors, we define the coefficients:

$$\zeta_i^\alpha := \theta^{A\alpha} \pi_{iA} \quad (i = 1, ..., n).$$

This allows us to express the Berezin integral identity:

$$\int d^8\theta \prod_{\alpha=1}^4 \zeta_i^\alpha \zeta_j^\alpha = \langle \pi_i, \pi_j \rangle^4. \tag{63}$$

Thus, the amplitude $\mathcal{A}_n$ can be reformulated as:

$$\mathcal{A}_n^{a_1...a_n} = ig^{n-2} \int \frac{d^4 X}{(2\pi)^4} \int d^8\theta \prod_{\alpha=1}^4 \zeta_1^\alpha \zeta_2^\alpha \mathrm{Tr}\left( \prod_{i=1}^n s_i^{e_i} e^{is_i q(z_i, \bar{z}_i) \cdot X} \frac{\mathsf{R}^{a_i}}{\langle \pi_i, \pi_{i+1} \rangle} \right). \tag{64}$$

This reformulation provides a compelling motivation for introducing the $\mathcal{N} = 4$ superfield $Z_i$, defined as follows:

$$Z_i = a^-(z_i, \bar{z}_i) + \zeta_i^\alpha \lambda_\alpha + \frac{1}{2} \zeta_i^\alpha \zeta_i^\beta \phi_{\alpha\beta} + \frac{1}{3!} \zeta_i^\alpha \zeta_i^\beta \zeta_i^\gamma \varepsilon_{\alpha\beta\gamma\delta} \tilde{\lambda}^\delta + a^+(z_i, \bar{z}_i) \prod_{\alpha=1}^4 \zeta_i^\alpha, \tag{65}$$

which describes the particle content of $\mathcal{N} = 4$ SYM theory. Here, the term $a_i^- := a^-(z_i, \bar{z}_i)$ corresponds to the vacuum expectation value of the annihilation operator for a negative-helicity gluon, while $a_i^+ := a^+(z_i, \bar{z}_i)$ represents the analogous quantity for a positive-helicity gluon. The dependence of the superfield $Z_i$ on the celestial sphere coordinates $(z_i, \bar{z}_i) \in \mathbf{CP}^1$ arises through the coefficients $\zeta_i^\alpha$.

Consequently, we introduce the (partial) super-amplitude as:

$$\mathbb{A}_n^{a_1...a_n} = ig^{n-2} \int \frac{d^4 X}{(2\pi)^4} \int d^8\theta \mathrm{Tr}\left( \prod_{i=1}^n s_i^{e_i} e^{is_i q(z_i, \bar{z}_i) \cdot X} \frac{\mathsf{R}^{a_i} Z_i}{\langle \pi_i, \pi_{i+1} \rangle} \right). \tag{66}$$

*Celestial Super-amplitude.* The celestial (partial) super-amplitude $\widehat{\mathbb{A}}_n^{a_1...a_n}$ is subsequently defined as the $\varepsilon$-regulated Mellin transform:

$$\widehat{\mathbb{A}}_n^{a_1...a_n} := \prod_{i=1}^n \int_{(0,\infty)} \frac{ds_i}{s_i} s_i^{\Delta_i} e^{-\varepsilon s_i} \mathbb{A}_n^{a_1...a_n}. \tag{67}$$

By performing the integral transform, $\widehat{\mathbb{A}}_n^{a_1...a_n}$ can be expressed in terms of the celestial wavefunctions $\phi_{2h_i}(z_i, \bar{z}_i | X)$ as:

$$\widehat{\mathbb{A}}_n^{a_1...a_n} = ig^{n-2} \int \frac{d^4 X}{(2\pi)^4} \int d^8\theta \mathrm{Tr}\left( \prod_{i=1}^n \phi_{2h_i}(z_i, \bar{z}_i | X) \frac{\mathsf{R}^{a_i} Z_i}{\langle \pi_i, \pi_{i+1} \rangle} \right), \tag{68}$$

where the scaling dimensions for the chosen MHV configuration are determined by the relations:

$$2h_i := \Delta_i + e_i, \ \ \forall\, 1 \le i \le n. \tag{69}$$

By employing the formalism of celestial leaf amplitudes, and under the simplifying assumption that all gluons are outgoing, the spacetime integral appearing in the preceding expression can be evaluated as follows:

$$\int d^4 X \prod_{i=1}^n \phi_{2h_i}\left(z_i, \bar{z}_i \big| X\right) = 2\pi\delta\left(\beta\right) \int_{\mathbf{H}_3} d^3x \prod_{i=1}^n G_{2h_i}\left(z_i, \bar{z}_i \big| x\right) + (\bar{z}_i \to -\bar{z}_i). \tag{70}$$

Here, the quantity $\beta$ is defined as:

$$\beta := 2\sum_{i=1}^n h_i - 4, \tag{71}$$

which represents the total scaling dimension associated with the scattering process in question.

Consequently, the celestial super-amplitude can be represented in the compact form:

$$\widehat{\mathbb{A}}_n^{a_1\dots a_n} = \frac{ig^{n-2}}{(2\pi)^3}\delta\left(\beta\right) A_n^{a_1\dots a_n} + (\bar{z}_i \to -\bar{z}_i), \tag{72}$$

where the leaf amplitude for gluons $A_n^{a_1\dots a_n}$ is defined by:

$$A_n^{a_1\dots a_n} := \int_{\mathbf{H}_3} d^3x \int d^8\theta \operatorname{Tr} \prod_{i=1}^n G_{2h_i}\left(z_i, \bar{z}_i \big| x\right) \frac{\mathsf{R}^{a_i} Z_i}{\langle \pi_i, \pi_{i+1}\rangle}. \tag{73}$$

*Remark.* Inspired by heuristic considerations analogous to those employed by Witten [38] in his formulation of twistor string theory, we find that the gluon leaf $n$-amplitude $A_n^{a_1\dots a_n}$ derived above admits the following representation as an integral over spinor variables:

$$A_n^{a_1\dots a_n} = \prod_{i=1}^n \int d^2\mu_i \frac{\Gamma\left(2h_i\right)}{(-i\mu_i \cdot \bar{\pi}_i)} \int_{\mathbf{H}_3} d^3x \int d^8\theta\, \phi^{a_1\dots a_n}, \tag{74}$$

where the mapping $(\mu_{\dot{A}i}, \pi_i^A) \in \mathbf{MT} \mapsto \phi^{a_1\dots a_n}$ on the $\mathcal{N}=4$ supersymmetric extension of minitwistor space $\mathbf{MT}$ (associated with Euclidean $AdS_3$) is defined as follows:

$$\phi^{a_1\dots a_n} = \operatorname{Tr} \prod_{i=1}^n \delta^{(2)}\left(\mu_{\dot{A}i} - \pi_i^A x_{A\dot{A}}\right) \frac{\mathsf{R}^{a_i} Z_i(\theta^{A\alpha}, \bar{\theta}_\alpha^{\dot{A}})}{\langle \pi_i, \pi_{i+1}\rangle}. \tag{75}$$

Here, the Grassmann-valued spinors $(\theta^{A\alpha}, \bar{\theta}_\alpha^{\dot{A}})$ appear in the arguments of the superfield $Z_i$ solely to emphasise how the dependence of $\phi^{a_1\dots a_n}$ on the supersymmetric extension of $\mathbf{MT}$ is introduced. Adopting the terminology of Cachazo and Skinner [3], Cachazo *et al.* [4], Adamo and Mason [5], we interpret $\phi^{a_1\dots a_n}$ as the multi-gluon wavefunction defined on minitwistor space. The set $\{(x^\mu, \theta^{A\alpha}, \bar{\theta}_\alpha^{\dot{A}}, \mu_{\dot{A}}, \pi^A)\}$ then represents the $\mathcal{N}=4$ supersymmetric extension of the configuration

space. Note that this supersymmetric configuration space is intimately related to the mathematical notion of *graded manifolds*, which were originally introduced in the study of dynamical systems possessing fermionic degrees of freedom. We refer the reader to the works of Kostant [39], Berezin [15] and Leites [16].

## B.   Euclidean $AdS_3$ String Theory and Gluon Leaf Amplitudes

Applying an analogous line of reasoning to that employed in the preceding section, we establish a correspondence between the gluon leaf amplitude $A_n$ and (Euclidean) $AdS_3$ string theory based on the following considerations. Recall from the prior discussion that, in the minisuperspace limit $k \to \infty$, the correlation functions of the primary fields $\Phi^{h_i}\left(w_i|z_i\right)$ in the level-$k$ $H_3^+$-WZNW model reduce to a contact Feynman-Witten diagram describing the propagation of massless particles on the hyperboloid $\mathbf{H}_3$. Explicitly, this is expressed as:

$$\lim_{k\to\infty} \prod_{i=1}^{n} \int d^2 w_i\, \Gamma\left(2h_i\right) \langle \Phi^{h_1}\left(w_1|z_1\right)...\Phi^{h_n}\left(w_n|z_n\right)\rangle = \int_{\mathbf{H}_3} d^3x \prod_{i=1}^{n} G_{2h_i}\left(z_i, \bar{z}_i|x\right). \qquad (76)$$

Here, $G_{2h_i}\left(z_i, \bar{z}_i|x\right)$ denotes the bulk-to-boundary Green's function associated with the covariant Laplacian on $\mathbf{H}_3$.

We now consider a solution to the equations of motion of (Euclidean) $AdS_3$ string theory, defined on the target space $AdS_3 \times \tilde{X}$, where $\tilde{X}$ is a compact manifold. Let $K^a\left(z\right)$ denote a level-one $SO(\tilde{N})$ WZNW current algebra defined on the boundary of (Euclidean) $AdS_3$, which is identified with the celestial sphere. We postulate that the currents $K^a\left(z\right)$ arises from the CFT defined on $\tilde{X}$.

Drawing upon the framework established by Giveon, Kutasov, and Seiberg [30], Dolan and Witten [31] and Dolan [32], it is known that, given a spinless operator $W_{\tilde{X}}$ in the $\tilde{X}$-CFT, physical vertex operators can be constructed in the form $W_{\tilde{X}}V_{jm\bar{m}}$, where $V_{jm\bar{m}}$ corresponds to the worldsheet vertex operators.

Building on this formalism, we define the *celestial gluon vertex operators* as:

$$\mathcal{U}_i^{a_i} := \int d^2 w_i\, \Gamma\left(2h_i\right) K^{a_i}(z_i)\Phi^{h_i}\left(w_i|z_i\right). \qquad (77)$$

The gluon leaf amplitude can thus be derived from the mini-superspace limit ($k \to \infty$) of Euclidean $AdS_3$ string theory and is expressed as:

$$A_n^{a_1...a_n} = \lim_{k\to\infty} \int d^8\theta \langle \mathcal{U}_1^{a_1}...\mathcal{U}_n^{a_n}\rangle. \qquad (78)$$

*Remark.* It is worth emphasising that the correspondence between celestial CFT and Euclidean $AdS_3$ string theory presented herein constitutes a significant refinement over the holographic dictionary proposed in our earlier work (Mol [2]). In the current formulation, celestial vertex operators for gluons and gravitons are treated on an equal footing. Specifically, both are constructed exclusively from quantities arising as solutions to the equations of motion of $AdS_3$ string theory. The distinction between these operators lies solely in their respective numbers of supersymmetry generators and the associated gauge groups.

## C.   Generating Functional

Employing reasoning analogous to that utilised in the preceding section, we construct a generating functional for the gluon leaf amplitudes, $A_n^{a_1 \cdots a_n}$, based on the chiral determinant representation of the WZNW action. The main distinction between the $\mathcal{N} = 8$ Supergravity and $\mathcal{N} = 4$ SYM cases lies in the number of supersymmetry generators and the structure of the respective gauge groups.

For the sake of clarity, we present a self-contained derivation rather than merely extrapolating the gravitational results to the gauge-theoretic case. While this approach entails some repetition, it serves to highlight the features of the gauge theory case. Nonetheless, it will become apparent that the fundamental structure of the generating functional remains invariant across both cases. This "universality" enables a systematic derivation of the action integral describing the proposed holographic $AdS_3$ theory.

Let $\tilde{\mathcal{S}}[G]$ denote the action functional for the level-one $SO(\tilde{N})$ WZNW model defined on $\mathbf{CP}^1$, where the dynamical variable is the matrix field $G = G(z, \bar{z})$, representing a mapping from $\mathbf{CP}^1$ to the Lie group $SO(\tilde{N})$. From $G$, we construct the differential one-form:

$$\mathscr{A} := \mathscr{A}_z dz + \mathscr{A}_{\bar{z}} d\bar{z}, \tag{79}$$

on $\mathbf{CP}^1$, with its holomorphic and anti-holomorphic components defined as follows:

$$\mathscr{A}_z = -(\partial_z G) G^{-1}, \;\; \mathscr{A}_{\bar{z}} = -(\partial_{\bar{z}} G) G^{-1}. \tag{80}$$

If $G$ satisfies the equations of motion of the WZNW model, the associated one-form $\mathscr{A}$ satisfies the "zero-curvature" condition:

$$\mathcal{F}[\mathscr{A}] := \partial_z \mathscr{A}_{\bar{z}} - \partial_{\bar{z}} \mathscr{A}_z + [\mathscr{A}_z, \mathscr{A}_{\bar{z}}] = 0. \tag{81}$$

Observe the analogy between this equality and the "flatness" condition $\mathcal{F}[\boldsymbol{\omega}] = 0$ encountered in the gravitational context. This parallel naturally suggests an interpretation of the differential one-form $\mathscr{A}$ as an $\mathfrak{so}(\tilde{N})$-valued connection one-form on $\mathbf{CP}^1$, associated with a gauge potential whose field strength is characterised by the curvature two-form $\mathcal{F}[\mathscr{A}]$. Consequently, we define the holomorphic component of the gauge-covariant derivative operator associated to the differential one-form $\mathscr{A}$ as:

$$\underline{\mathrm{d}}_z := \partial_z + [\mathscr{A}_z, \cdot]. \tag{82}$$

As reviewed by Nair [33], the action functional $\mathcal{S}[G]$ for the level-one $SO(\tilde{N})$ WZNW model on $\mathbf{CP}^1$ can be expressed in terms of the chiral Dirac determinant as follows:

$$\tilde{\mathcal{S}}[G] =: \tilde{\mathcal{S}}[\mathscr{A}] = \mathrm{Tr}\log\underline{\mathrm{d}}_z - \mathrm{Tr}\log\partial_z. \tag{83}$$

This expression admits a formal expansion in terms of the components $\mathscr{A}_z^{a_i}(z_i, \bar{z}_i)$:

$$\tilde{\mathcal{S}}[\mathscr{A}] = \sum_{m\geq 2} \frac{(-1)^{m+1}}{m} \int \frac{d^2 z_1}{\pi}...\int \frac{d^2 z_m}{\pi} \mathrm{Tr}\left(\frac{\mathscr{A}_z^{a_1}(z_1, \bar{z}_1)...\mathscr{A}_z^{a_n}(z_m, \bar{z}_m)}{z_{12}z_{23}...z_{m1}}\right),$$

where $a_i$ represents the colour indices associated with the $SO(\tilde{N})$ gauge group. The primary distinction between this formulation and the analogous expression for $\mathcal{S}[\boldsymbol{\omega}]$ in the gravitational case lies in the structure of the colour indices.

As in the preceding analysis, we consider the possibility of lifting the gauge potential $\mathscr{A}$ from $\mathbf{CP}^1$ to $\tilde{\mathscr{A}}$ defined on the $\mathcal{N} = 4$ supersymmetric extension of $\mathbf{H}_3$. The holomorphic component of the extended potential, $\tilde{\mathscr{A}}_z^{a_i}$, is expressed as:

$$\tilde{\mathscr{A}}_z^{a_i}(z_i, \bar{z}_i) = \int_{\mathcal{P}} \frac{d\Delta_i}{2\pi i} G_{2h_i}(z_i, \bar{z}_i | x) \mathsf{R}^{a_i} Z_i. \tag{84}$$

Here, $\tilde{\mathscr{A}}_z^{a_i}$ differs from $\boldsymbol{\Omega}_i$, introduced in the gravitational case, in two key respects: the number of super-symmetries, contained in the superfield $Z_i$, and the gauge structure, described by the $\mathsf{R}^a$ generators. Consequently, we define the "effective" action functional for the gluon leaf amplitudes as follows:

$$\exp(\mathcal{W}_{YM}[\tilde{\mathscr{A}}]) := \int_{\mathbf{H}_3} d^3 x \int d^8 \theta \tilde{\mathcal{S}}[\tilde{\mathscr{A}}_z^{a_i}]. \tag{85}$$

By taking functional derivatives of $e^{\mathcal{W}_{YM}}$ with respect to $a_i^{\pm a_i}$, representing the vacuum expectation values for the annihilation operators of gluons with positive and negative helicities, we obtain the gluon leaf amplitude:

$$A_n = \frac{\delta}{\delta a_1^-} \frac{\delta}{\delta a_2^-} \frac{\delta}{\delta a_3^+}...\frac{\delta}{\delta a_n^+} \exp(\mathcal{W}_{YM}[\tilde{\mathscr{A}}])\Big|_{\tilde{\mathscr{A}}=0}. \tag{86}$$

This construction demonstrates that the "effective" action $e^{\mathcal{W}_{YM}}$ serves as the generating functional for the gluon leaf amplitudes, as previously asserted. Furthermore, as noted at the outset of this section, the principal differences between the gravitational and gauge-theoretic cases arises from the number of degrees of freedom contained in the respective superfields ($\Psi_i$ *versus* $Z_i$) and the structure of the Lie group ($\mathsf{T}^a$ generators *versus* $\mathsf{R}^a$ generators).

Despite these distinctions, the fundamental structure of the generating functional remains invariant. This "universality" enables a systematic and unified analysis of the action functional governing the holographic $AdS_3$ theory for both gravitons and gluons.

## IV. HOLOGRAPHIC $AdS_3$ MODELS

### A. Introductory Remarks

To explain our approach to deriving the action integral for the WZNW-like field theory on $AdS_3$ that generates the leaf amplitudes for gravitons and gluons, we begin by briefly reviewing, in accordance with Weinberg [40], the formulation of the $S$-matrix in terms of the effective action.

Let $\varphi^I(x)$ denote a set of fields indexed by $I$, and let $\Gamma[\varphi^I(x)]$ represent the effective quantum action. The quantum equations of motion are determined by the stationarity points of $\Gamma[\varphi^I(x)]$, satisfying the condition:

$$\frac{\delta}{\delta \varphi^I} \Gamma = 0. \tag{87}$$

The generating functional that yields the $S$-matrix elements is then defined as:

$$F := \exp(i\Gamma[\varphi^I])\Big|_{\frac{\delta\Gamma}{\delta\varphi^I}=0}. \tag{88}$$

In perturbative quantum field theory, the solutions to the equations of motion are expanded around the free field configuration:

$$\varphi^I(x) \propto \int dk\, [a(k)u_k(x) + a^*(k)u_k^*(k)], \tag{89}$$

where $u_k(x)$ are plane-wave modes, and $a(k)$, $a^*(k)$ denote the mode coefficients. The scattering amplitude for a process of the form $p_1, p_2...p_n \to p_1', p_2'...p_m'$ is computed using the expression:

$$\mathcal{A} = \left( \prod_{i=1}^n \frac{\delta}{\delta a(p_i)} \prod_{j=1}^n \frac{\delta}{\delta a(p_j')} F \right)_{a(k)=a^*(k)=0}. \tag{90}$$

This expression coincides formally with the graviton leaf amplitude $M_n$ and the gluon leaf amplitude $A_n^{a_1...a_n}$ obtained in Eqs. (59) and (86). Therefore, the next level of abstraction is

introduced as follows: we *postulate* that the classical *effective* actions $\mathcal{W}_{SUGRA}[\mathbf{\Omega}]$ and $\mathcal{W}_{YM}[\tilde{\mathscr{A}}]$, which govern the holographic $AdS_3$ theories, take the following forms:

$$\mathcal{W}_{SUGRA}[\mathbf{\Omega}] = \int_{\mathbf{H}_3} d^3x \int d^{16}\theta \, \mathcal{S}[\mathbf{\Omega}], \tag{91}$$

and:

$$\mathcal{W}_{YM}[\tilde{\mathscr{A}}] := \int_{\mathbf{H}_3} d^3x \int d^8\theta \tilde{\mathcal{S}}[\tilde{\mathscr{A}}_z^{a_i}], \tag{92}$$

which correspond, respectively, to the dual descriptions of the MHV subsector of celestial CFT for $\mathcal{N} = 8$ Supergravity and $\mathcal{N} = 4$ SYM theory.

Under these assumptions, the results derived in Sections II and III, specifically Eqs. (59) and (86), are precisely reproduced as particular instances of Eq. (90) for the respective actions. Explicitly, these amplitudes are expressed as:

$$M_n = \left( \prod_{i=1}^{n} \frac{\delta}{\delta h^{\sigma_i}(z_i, \bar{z}_i)} F_{SUGRA} \right)_{\mathbf{\Omega}=0}, \tag{93}$$

and:

$$A_n = \left( \prod_{i=1}^{n} \frac{\delta}{\delta a^{\bar{\sigma}_i}(z_i, \bar{z}_i)} F_{YM} \right)_{\tilde{\mathscr{A}}=0}, \tag{94}$$

where the generating functionals are defined as:

$$F_{SUGRA} := \exp(\mathcal{W}_{SUGRA}[\mathbf{\Omega}]), \tag{95}$$

and:

$$F_{YM} := \exp(\mathcal{W}_{YM}[\tilde{\mathscr{A}}]). \tag{96}$$

In these formulae, $\sigma_i$ and $\bar{\sigma}_i$ denote the helicities of gravitons and gluons, respectively, while $h^{\sigma_i}(z_i, \bar{z}_i)$ and $a^{\bar{\sigma}_i}(z_i, \bar{z}_i)$ represent the expectation values of the annihilation operators for gravitons and gluons. These operators are contained within the superfields $\Psi_i$ (for $\mathcal{N} = 8$ Supergravity) and $Z_i$ (for $\mathcal{N} = 4$ SYM theory), respectively.

We now delineate the framework for constructing holographic $AdS_3$ theories. Abe, Nair, and Park [8] and Abe [9] demonstrated that the superspace constraints of $\mathcal{N} = 4$ SYM theory and the anti-self-dual sector of $\mathcal{N} = 8$ Supergravity can be embedded in a supersymmetric extension of twistor space. Within this formalism, it was further shown by these authors that one can impose the so-called *chiral semi-analytic gauge* on the constraint equations, adopting the terminology

introduced in the harmonic superspace literature, as comprehensively revised by Galperin *et al.* [10]. From these results, we extract two important corollaries that inform our subsequent analysis.

The first corollary asserts that, upon imposing the chiral semi-analytic gauge, the residual constraint is precisely identified with the zero-curvature conditions $\mathcal{F}[\boldsymbol{\omega}] = 0$ and $\mathcal{F}[\mathscr{A}] = 0$, as derived in Sections II and III for $\mathcal{N} = 8$ Supergravity and $\mathcal{N} = 4$ SYM theory, respectively. These zero-curvature conditions will emerge as the Euler-Lagrange equations of the proposed action for the holographic $AdS_3$ models.

The second corollary pertains to the geometric interpretation of the celestial leaf amplitudes. As established in Sections II and III, multi-graviton and multi-gluon celestial leaf amplitudes naturally admit a representation within minitwistor space. Consequently, solving the so-called *analyticity condition* for the gauge superpotentials in the supersymmetric twistor space, and subsequently performing a scaling reduction from twistor to minitwistor space, following the approach of Bu and Seet [11], yields the lifted potentials $\boldsymbol{\Omega}$ and $\tilde{\mathscr{A}}$. These potentials were introduced in Eqs. (56) and (84) of Sections II and III, respectively, and are essential for constructing the generating functionals for celestial leaf amplitudes. For completeness, we recall that the holomorphic components of these potentials are explicitly given by:

$$\Omega_z(z_i, \bar{z}_i | x, \theta, \bar{\theta}) := \int_{\mathcal{P}} \frac{d\Delta_i}{2\pi i} G_{2h_i}\left(z_i, \bar{z}_i | x\right) \Psi_i, \tag{97}$$

and:

$$\tilde{\mathscr{A}}_z^{a_i}(z_i, \bar{z}_i) = \int_{\mathcal{P}} \frac{d\Delta_i}{2\pi i} G_{2h_i}\left(z_i, \bar{z}_i | x\right) \mathsf{R}^{a_i} Z_i. \tag{98}$$

Finally, we will propose an action integral for the holographic WZNW-like field theories on $AdS_3$, demonstrate that its Euler-Lagrange equations recover the superspace constraints in the chiral semi-analytic gauge (or equivalently, the zero-curvature conditions, $\mathcal{F}[\boldsymbol{\omega}] = 0$ and $\mathcal{F}[\mathscr{A}] = 0$) and conclude by showing that the corresponding on-shell effective actions yield Eqs. (59) and (86).

### B. Holomorphic Superpotential

For the purposes of our present analysis, the main lessons from Abe, Nair, and Park [8] and Abe [9] can be summarised as follows. The superspaces associated with $\mathcal{N} = 4$ SYM theory and the anti-self-dual sector of $\mathcal{N} = 8$ Supergravity may be extended to include a pair of two-component spinors, $(\mu^A, \nu_{\dot{A}})$, together with an auxiliary frame vector $W_{A\dot{A}}$, related through the equation:

$$\nu_{\dot{A}} = \mu^A W_{A\dot{A}}. \tag{99}$$

In what follows, this extension of superspace will be interpreted *heuristically* as a supersymmetric generalisation of twistor space. For a mathematically rigorous account of the construction of twistor superspace, the reader is referred to the original works of Lukierski [41], Lukierski and Nowicki [42], Kotrla and Niederle [43]. For recent developments, see Popov and Saemann [44] and Wolf [45].

Within this framework, Abe, Nair, and Park [8] and Abe [9] established that the constraints of $\mathcal{N} = 4$ SYM theory and the anti-self-dual sector of $\mathcal{N} = 8$ Supergravity can be formulated in the so-called *chiral semi-analytic gauge*, a terminology originating in the harmonic superspace literature, as reviewed by Galperin *et al.* [10], Ivanov [46]. In this gauge, the superpotentials, denoted $H^{++}$ and $H^{--}$, satisfy the following system of constraint equations:

$$\overline{D}^{\alpha}_{\dot{A}} H^{++} = 0, \quad \text{(chirality condition)} \tag{100}$$

$$D^{(+)}_{\alpha} H^{++} = 0 \quad \text{(analyticity condition)}, \tag{101}$$

$$D^{(++)} H^{--} - D^{(--)} H^{++} + [H^{++}, H^{--}] = 0, \tag{102}$$

where the derivative operators are defined as follows:

$$D_{A\alpha} = \frac{\partial}{\partial \theta^{A\alpha}} + i(\sigma^{\mu})_{A\dot{A}} \bar{\theta}^{\dot{A}}_{\alpha} \frac{\partial}{\partial x^{\mu}} \quad \overline{D}^{\alpha}_{\dot{A}} := -\frac{\partial}{\partial \bar{\theta}^{\dot{A}}_{\alpha}} - i\theta^{A\alpha}(\sigma^{\mu})_{A\dot{A}} \frac{\partial}{\partial x^{\mu}}, \tag{103}$$

$$D^{(+)}_{\alpha} := \mu^{A} D_{A\alpha}, \quad D^{(++)} := \mu^{A} \frac{\partial}{\partial \bar{\nu}^{A}}, \quad D^{(--)} := -\bar{\nu}^{A} \frac{\partial}{\partial \mu^{A}}. \tag{104}$$

A concise definition of the gauge superpotentials $H^{++}$ and $H^{--}$ is provided by Abe, Nair, and Park [8] and Abe [9]. For a rigorous mathematical construction of $H^{++}$ and $H^{--}$, and the "harmonic derivatives" $D^{(+)}_{\alpha}$, $D^{(++)}$ and $D^{(--)}$, as well as their gauge-theoretic properties, we refer the reader to Galperin *et al.* [10].

To summarise, the superspace constraints of $\mathcal{N} = 4$ SYM theory and the anti-self-dual sector of $\mathcal{N} = 8$ Supergravity (which, as defined by Abe [9], is the sector of vanishing spinorial curvature, $R^{ab}_{\alpha\beta} = 0$), when expressed in the chiral semi-analytic gauge within this twistor-like extension of superspace, exhibit an *identical structural form*. The above constraint equations for these theories differ solely in the number of *supersymmetry generators* associated with the superpotentials and their respective *gauge groups*. Accordingly, we employ the notation $H^{++}, H^{--}$ to refer generically to the gauge superpotentials for either of these theories. In what follows, we adopt an abstract and generic approach, ensuring that our results are applicable to both $\mathcal{N} = 4$ SYM theory and the anti-self-dual sector of $\mathcal{N} = 8$ Supergravity.

### 1.  Local Parametrisation

We now proceed to reformulate Eq. (102) by introducing a local parametrisation for the pair $(\mu^A, \nu_{\dot{A}})$ of two-component spinors, alongside the four-vector $W_{A\dot{A}}$. Let us write $\mu^A = (\xi, \zeta)^T$ and assume the existence of an open neighbourhood $\mathcal{U}$ where $\zeta \neq 0$. In this region, the local coordinate representation of $\mu^A$ on the complex projective line can be defined by $z := \xi/\zeta$. Furthermore, we choose a convenient frame wherein the four-vector is fixed to $W_{A\dot{A}} = \delta_{A\dot{A}}$. Under this parametrisation, the spinors $\mu^A$ and $\nu_{\dot{A}}$ can be identified, respectively, with $\mu^A = \pi^A = (z, 1)^T$ and $\nu_{\dot{A}} = \bar{\pi}_{\dot{A}} = (\bar{z}, -1)^T$ (up to little-group rescaling), which were previously employed in the analysis of graviton and gluon celestial amplitudes in Sections II and III.

We now introduce a local parametrisation of the gauge superpotentials, defined on the neighbourhood $\mathcal{U}$ as follows:

$$(H^{(++)})_{\mathcal{U}} = \xi\bar{\xi}^{-1}(1 + z\bar{z})H_z, \quad (H^{(--)})_{\mathcal{U}} = \bar{\xi}\xi^{-1}(1 + z\bar{z})H_{\bar{z}}. \tag{105}$$

In this representation, the constraint expressed by Eq. (102) assumes the form:

$$\partial_z H_{\bar{z}} - \partial_{\bar{z}} H_z + [H_z, H_{\bar{z}}] = 0. \tag{106}$$

This equation is thereby identified with the zero-curvature conditions $\mathcal{F}[\boldsymbol{\omega}] = 0$ and $\mathcal{F}[\mathscr{A}] = 0$, previously derived in Sections II and III in the contexts of $\mathcal{N} = 8$ Supergravity and $\mathcal{N} = 4$ SYM theory, respectively.

Henceforth, we *formally* define $H := H_z dz + H_{\bar{z}} d\bar{z}$ and designate $H$ as *the* superpotential, with its holomorphic and anti-holomorphic components represented by $H_z$ and $H_{\bar{z}}$, respectively. Observe the structural analogy between the superpotential $H$ thus defined and the $\mathbf{CP}^1$ differential one-forms $\boldsymbol{\omega} = \omega_z dz + \omega_{\bar{z}} d\bar{z}$ (cf. Eq. (51)) and $\mathscr{A} = \mathscr{A}_z dz + \mathscr{A}_{\bar{z}} d\bar{z}$ (cf. Eq. (79)), as introduced in Sections II and III, respectively. Our objective is to show that this correspondence *extends beyond a merely formal analogy*.

### 2.  Fourier Representation and Analyticity

To address the analyticity condition, Eq. (101), we first examine the implications of the chirality constraint, Eq. (100), on the Fourier representation of the holomorphic component $H_z$. This constraint necessitates that $H$ depends on the Grassmann-valued two-component spinors $\theta^{A\alpha}$ and $\bar{\theta}_{\alpha}^{\dot{A}}$ exclusively through the combination:

$$Y^\mu(\mathbb{X}) := X^\mu + i\theta^{A\alpha}(\sigma^\mu)_{A\dot{A}}\bar{\theta}_{\alpha}^{\dot{A}}, \tag{107}$$

where $\mathbb{X}^I := (X^\mu, \theta^{A\alpha}, \bar{\theta}^{\dot{A}}_{\dot{\alpha}})$ denotes the coordinates of the $\mathcal{N}$-extended supersymmetric flat space.

Following Weinberg [40], the holomorphic component $H_z = H_z(\mathbb{X})$ of the superpotential is then expressed in terms of an on-shell Fourier decomposition involving the mode functions $\boldsymbol{\alpha}^a_\sigma(p)$:

$$H_z^a = \sum_\sigma \int d^4p \; \theta\left(p^0\right) \delta(p^2) e^{ip \cdot Y} \boldsymbol{\alpha}^a_\sigma(p). \tag{108}$$

Here, $\sigma$ represents the helicity of the one-particle irreducible representations of the Lorentz group, associated with the massless gauge bosons of the theory (cf. Weinberg [47]), while $a$ denotes the colour index corresponding to the gauge group. For notational economy, we restrict our attention to the positive-frequency modes, enforced through $\theta(p^0)$, noting that the extension to include negative-frequency modes is straighforward.

*Momentum-space Parametrisation.* As the Fourier decomposition in Eq. (108) is explicitly on-shell, enforced by the delta function $\delta(p^2)$, the integration is restricted to the null cone in momentum space, $\{p^\mu | p^2 = 0\}$. To facilitate further analysis, we perform a change of integration variables, mapping from Cartesian momentum-space coordinates $p^\mu$ to a parametrisation in terms of the frequency $s$ and celestial sphere coordinates $z, \bar{z}$, given by:

$$p^\mu(s, z, \bar{z}) = sq^\mu(z, \bar{z}) = s\left(1 + z\bar{z}, z + \bar{z}, i\left(\bar{z} - z\right), 1 - z\bar{z}\right). \tag{109}$$

In this parametrisation, the Lorentz-invariant measure transforms as:

$$\frac{d^3\vec{p}}{2p^0} = ds d^2z \, is. \tag{110}$$

Substituting this change of variables, the expression for $H_z^a$ becomes:

$$H_z^a = \sum_\sigma \int_{\mathbf{R}_+^\times} \frac{ds}{s} \int_{\mathbf{C}} d^2z \, s^2 e^{isq(z,\bar{z}) \cdot Y(\mathbb{X})} \boldsymbol{\alpha}^a_\sigma(sq(z, \bar{z})). \tag{111}$$

Here, $\mathbf{R}_+^\times$ denotes the multiplicative group of positive reals, with $ds/s$ representing the corresponding Haar measure.

*The Holomorphic Potential Density.* Following Pasterski and Shao [23], we recall the Mellin-integral identity:

$$e^{isq(z,\bar{z}) \cdot Y(\mathbb{X})} = \int_{\mathcal{P}} \frac{d\Delta}{2\pi i} s^{-\Delta} \frac{\Gamma(\Delta)}{(\varepsilon - iq(z, \bar{z}) \cdot Y)^\Delta}, \tag{112}$$

where $\mathcal{P} := 1 + i\mathbf{R}$. Define the Mellin-transformed Fourier modes $\hat{\boldsymbol{\alpha}}^a_\sigma = \hat{\boldsymbol{\alpha}}^a_\sigma(\Delta, z, \bar{z})$ by:

$$\hat{\boldsymbol{\alpha}}^a_\sigma := i \int_{\mathbf{R}_+^\times} \frac{ds}{s} \; s^{2-\Delta} \boldsymbol{\alpha}^a_\sigma(sq(z, \bar{z})). \tag{113}$$

With this definition, the holomorphic component of the superpotential $H_z^a$ can equivalently be expressed as:

$$H_z^a = \sum_\sigma \int_{\mathbf{C}} d^2z\, h_\sigma^a(z, \bar{z}|\mathbb{X}), \tag{114}$$

where the *holomorphic superpotential density* $h_\sigma^a = h_\sigma^a(z, \bar{z}|\mathbb{X})$ is given by:

$$h_\sigma^a\left(z, \bar{z}|\mathbb{X}\right) := \int_{\mathcal{P}} \frac{d\Delta}{2\pi i} \tilde{\phi}_\Delta(z, \bar{z}|\mathbb{X}) \hat{\boldsymbol{\alpha}}_\sigma^a(\Delta, z, \bar{z}). \tag{115}$$

Here, $\tilde{\phi}_\Delta$ represents the chiral supersymmetric extension of the celestial conformal primary basis for massless scalars, defined as:

$$\tilde{\phi}_\Delta(z, \bar{z}|\mathbb{X}) := \frac{\Gamma(\Delta)}{\left(\varepsilon - iq\left(z, \bar{z}\right) \cdot X + \xi^\alpha \bar{\xi}_\alpha\right)}, \tag{116}$$

where the coefficients:

$$\xi^\alpha := \theta_A^\alpha \pi^A, \quad \bar{\xi}_\alpha := \bar{\theta}_\alpha^{\dot{A}} \bar{\pi}_{\dot{A}} \quad (\alpha = 1, ..., \mathcal{N}), \tag{117}$$

are as defined in Sections II and III. Note that $h_\sigma^a$ is a density on the celestial sphere.

Employing the definitions of the superspace spinor and harmonic derivatives from Eqs. (103) and (104), it follows that $\tilde{\phi}_\Delta$ satisfies the following important properties:

$$\overline{D}_{\dot{A}}^\alpha \tilde{\phi}_\Delta(z, \bar{z}|\mathbb{X}) = 0 \ \text{ (chirality condition)}, \tag{118}$$

$$D_\alpha^{(+)} \tilde{\phi}_\Delta(z, \bar{z}|\mathbb{X}) = 0 \ \text{ (analyticity condition)}. \tag{119}$$

*Analyticity.* We now derive the implications of the analyticity constraint imposed on the superpotential. Utilising Eq. (119), together with the Fourier representation, we find:

$$D_i^{(+)} h_\sigma^a = \int_{\mathscr{C}} \frac{d\Delta}{2\pi i} \tilde{\phi}_\Delta(z, \bar{z}|\mathbb{X}) \pi^A \frac{\partial \hat{\boldsymbol{\alpha}}_\sigma^a(\Delta, z, \bar{z})}{\partial \theta^{Ai}}. \tag{120}$$

The analyticity condition on the superpotential then requires that:

$$\pi^A \frac{\partial \hat{\boldsymbol{\alpha}}_\sigma^a(\Delta, z, \bar{z})}{\partial \theta^{Ai}} = 0. \tag{121}$$

Since $\hat{\boldsymbol{\alpha}}_\sigma^a$ depends on $\theta^{A\alpha}$ exclusively through the coefficients $\xi^\alpha$, it follows that the Mellin-transformed mode functions must be of the form:

$$\hat{\boldsymbol{\alpha}}_\sigma^a(\Delta, z, \bar{z}) = W_{\sigma,\Delta}(z, \bar{z})\mathsf{S}^a = \left(\alpha_- + \xi^\alpha \lambda_\alpha + ... + \alpha_+ \prod_{\alpha=1}^{\mathcal{N}} \xi^\alpha\right) \mathsf{S}^a, \tag{122}$$

where $W_\sigma$ is the superfield describing the particle multiplet of the theory under consideration. Here, $\alpha_- = \alpha_-(\Delta, z, \bar{z})$ denotes the Mellin-transformed expectation value of the annihilation operator for a massless boson with negative helicity, whereas $\alpha_+ = \alpha_+(\Delta, z, \bar{z})$ corresponds to the analogous quantity for positive helicity.

Thus, we have recovered the superfields $\Psi$ (for $\mathcal{N} = 8$ Supergravity) and $Z$ (for $\mathcal{N} = 4$ SYM theory). Consequently, the holomorphic potential density takes the form:

$$h_\sigma^a\left(z, \bar{z}\big|\mathbb{X}\right) := \int_\mathcal{P} \frac{d\Delta}{2\pi i} \tilde{\phi}_\Delta(z, \bar{z}\big|\mathbb{X}) W_{\sigma,\Delta}(z, \bar{z}) \mathsf{S}^a. \tag{123}$$

*Scaling Reduction.* We now proceed to motivate the scaling reduction from twistor to minitwistor space. It is pertinent to recall that our application of twistor theory serves primarily as a heuristic guide.

The expression for $h_\sigma^a$ can be reformulated as an integral over a two-component spinor, as follows:

$$h_\sigma^a\left(z, \bar{z}\big|\mathbb{X}\right) := \int_\mathcal{P} \frac{d\Delta}{2\pi i} \int d^2\mu \, \frac{\Gamma(\Delta)}{\left(-i\mu \cdot \bar{\pi} + \xi^\alpha \bar{\xi}_\alpha\right)} f_{\sigma,\Delta}^a, \tag{124}$$

where $f_{\sigma,\Delta}^a$ is defined on the holomorphically extended twistor space as:

$$f_{\sigma,\Delta}^a(z, \bar{z}\big|X, \mu_{\dot{A}}, \pi^A) := \delta^{(2)}(\mu_{\dot{A}} - \pi^A X_{A\dot{A}}) W_{\sigma,\Delta}(z, \bar{z}) \mathsf{S}^a. \tag{125}$$

Here, the equality $\mu_{\dot{A}} - \pi^A X_{A\dot{A}} = 0$ corresponds to the twistor equation, as explained in Appendix A of Witten [38]. From Sections II and III, we have learned that the multi-graviton and multi-gluon celestial amplitudes in a MHV configuration admit a natural geometric interpretation within the minitwistor space $\mathbf{MT}$ of Euclidean $AdS_3$.

Following Bu and Seet [11], the scaling reduction is now performed by identifying two-component spinors $\mu_{\dot{A}} \simeq \mu'_{\dot{A}}$ whenever there exists a *complex* scalar $a \in \mathbf{C}$ such that $\mu_{\dot{A}} = a\mu'_{\dot{A}}$. Under this equivalence, the scale-reduced counterpart of $f_{\sigma,\Delta}^a$ is given by:

$$\tilde{f}_{\sigma,\Delta}^a(z, \bar{z}\big|x^\mu, \mu_{\dot{A}}, \pi^A) := \delta^{(2)}(\mu_{\dot{A}} - \pi^A x_{A\dot{A}}) W_{\sigma,\Delta}(z, \bar{z}) \mathsf{S}^a, \tag{126}$$

where $x_{A\dot{A}} \in \mathbf{H}_3$ is a point on the hyperboloid and $(\mu_{\dot{A}}, \pi^A) \in \mathbf{MT}$ is a point on the minitwistor space of $\mathbf{H}_3$.

Accordingly, the scale-reduced holomorphic potential density takes the form:

$$\tilde{h}_{\sigma,\Delta}^a(z, \bar{z}\big|x^\mu, \mu_{\dot{A}}, \pi^A) = \int_\mathcal{P} \frac{d\Delta}{2\pi i} \tilde{G}_\Delta(z, \bar{z}\big|\mathbb{X}) W_{\sigma,\Delta}(z, \bar{z}) \mathsf{S}^a, \tag{127}$$

where $\tilde{G}_\Delta$ denotes the chiral supersymmetric extension of the bulk-to-boundary Green's function on the $\mathcal{N}$-supersymmetric hyperboloid $\mathbf{H}_3$. Thus, we recovered the lifted potentials $\mathbf{\Omega}$ for $\mathcal{N} = 8$ Supergravity, introduced in Section II, and $\mathscr{A}$ for $\mathcal{N} = 4$ SYM theory, introduced in Section III, as we wished.

*Remark.* From a mathematical perspective, the geometric structure underlying our holographic models is that of a holomorphic line bundle, wherein the complex projective line $\mathbf{CP}^1$ serves as the fibre over the $\mathcal{N}$-extended supersymmetric $AdS_3$. The superpotential $\tilde{h}^a_{\sigma,\Delta}$ is naturally identified as a section of this fibre bundle.

## C. $AdS_3$ Action Integral

In this concluding section, we propose the action integral governing the holographic $AdS_3$ models, which reproduces the tree-level MHV celestial leaf super-amplitudes. Let $\mathcal{N}$ denote the number of supersymmetry generators, and let $\mathcal{G}$ represent the gauge group associated with special orthogonal matrices. For notational simplicity, we define:

$$\int d\mathbb{X} \, (\cdot) := \int_{\mathbf{H}_3} d^3 x \int d^{\mathcal{N}} \theta \, (\cdot). \tag{128}$$

Let $\mathcal{S}[G]$ denote the action functional for the level-one $\mathcal{G}$ WZNW model on $\mathbf{CP}^1$. We postulate the following action for the holographic $AdS_3$ models:

$$\mathcal{I}[G, H, e] := - \int_{\mathbf{H}_3} d\mathbb{X} \, \mathcal{S}[G] + \Theta[G, H, e], \tag{129}$$

where the term $\Theta[G, H, e]$ is defined by:

$$\Theta[G, H, e] := \frac{1}{\pi} \int d\mathbb{X} \int d^2 z \, \left( \mathrm{Tr} \left[ H_z \left( \partial_{\bar{z}} G \right) G^{-1} \right] + e(H_{\bar{z}} + \left( \partial_{\bar{z}} G \right) G^{-1}) \right). \tag{130}$$

In this expression, $\mathrm{Tr}(\cdot)$ denotes the trace taken over the matrix Lie group $\mathcal{G}$, and $e$ is a Lagrange multiplier.

### 1. *Euler-Lagrange Equations*

The variation of Eq. (129) with respect to the Lagrange multiplier $e$ trivially yields the constraint equation:

$$H_{\bar{z}} + (\partial_{\bar{z}} G) G^{-1} = 0. \tag{131}$$

To compute the variation of the WZNW action $\mathcal{S}[G]$, we apply the Polyakov-Wiegmann identity, originally derived in Polyakov and Wiegmann [48]. For two $\mathcal{G}$-valued matrix fields $M = M(z, \bar{z})$ and $N = N(z, \bar{z})$, the identity reads:

$$\mathcal{S}[MN] = \mathcal{S}[M] + \mathcal{S}[N] - \frac{1}{\pi} \int d^2 z \, [\mathrm{Tr} M^{-1} (\partial_{\bar{z}} M)(\partial_z N) N^{-1}]. \tag{132}$$

Let $\|A\| := (\mathrm{Tr} A^T A)^{1/2}$ define the norm on the matrix Lie group $\mathcal{G}$, and let $\lambda$ denote an "infinitesimal" matrix field. Using the Polyakov-Wiegmann identity, we compute the variation:

$$\mathcal{S}[G^T(1+\lambda)] - \mathcal{S}[G^T] = \frac{1}{\pi} \int d^2z \, \mathrm{Tr}[\partial_z(G^{T-1}(\partial_{\bar{z}} G^T)\lambda)] + \mathcal{O}(\|\lambda\|^2). \tag{133}$$

By setting $\lambda = G^{T-1} \delta G^T$, the variation of $\mathcal{S}$ with respect to $G^T$ becomes:

$$\delta_{G^T} \mathcal{S} = \frac{1}{\pi} \int d^2z \, \mathrm{Tr}[\partial_z(G^{T-1}(\partial_{\bar{z}} G^T))G^{T-1}\delta G^T] + \mathcal{O}(\|\lambda\|^2). \tag{134}$$

The next step is to evaluate the variation of the term $\Theta$ with respect to $G^T$, under the constraint $H_{\bar{z}} + (\partial_{\bar{z}} G)G^{-1} = 0$. Rewriting $\Theta$ in terms of $G^T$, we have:

$$(\Theta[G, H, e])_{\frac{\delta \mathcal{I}}{\delta e}=0} = -\frac{1}{\pi} \int d\mathbb{X} \int d^2z \, \mathrm{Tr}(H_z G^{T-1}(\partial_{\bar{z}} G^T)). \tag{135}$$

The variation $\delta_{G^T} \Theta$ is the computed as:

$$\delta_{G^T} \Theta = \frac{1}{\pi} \int d\mathbb{X} \int d^2z \, \mathrm{Tr}[(G^{T-1}(\partial_{\bar{z}} G^T)H_z G^{T-1} + \partial_{\bar{z}}(H_z G^{T-1}))\delta G^T]. \tag{136}$$

Therefore, the equation of motion obtained from the variation of $\mathcal{I}$ with respect to $G^T$ becomes:

$$-\partial_z((\partial_{\bar{z}} G)G^{-1}) = -(\partial_{\bar{z}} G)G^{-1}H_z + \partial_{\bar{z}} H_z + H_z(\partial_{\bar{z}} G)G^{-1}. \tag{137}$$

Substituting the constraint $H_{\bar{z}} + (\partial_{\bar{z}} G)G^{-1} = 0$, we find:

$$\partial_z H_{\bar{z}} = \partial_{\bar{z}} H_z + [H_{\bar{z}}, H_z] = 0, \tag{138}$$

which precisely corresponds to the "zero-curvature" condition identified by Abe, Nair, and Park [8] and Abe [9] as the residual superspace constraint equations in the chiral semi-analytic gauge.

## 2. The On-shell Effective Action

We now demonstrate that the on-shell effective action $\mathcal{W}_{AdS_3}$ derived from $\mathcal{I}$ is the generating function for the leaf amplitudes.

To begin, consider a $\mathcal{G}$-valued matrix field $M = M(z, \bar{z})$ satisfying $H_z = M^T \partial_z M^{T-1}$. Substituting this decomposition of the holomorphic superpotential into the equation of motion obtained above, we deduce that $M = G^T$. Thus, the solution $H = H_z dz + H_{\bar{z}} d\bar{z}$ to the equations of motion, which is equivalently a solution to the superspace constraints in the chiral semi-analytic gauge, is entirely characterised by $G$, with

$$H_z = -(\partial_z G)G^{-1}, \quad H_{\bar{z}} = -(\partial_{\bar{z}} G)G^{-1}. \tag{139}$$

Consequently, the functional $\Theta$, evaluated on-shell with $\delta\mathcal{I}/\delta G = \delta\mathcal{I}/\delta e = 0$, becomes:

$$(\Theta[G, H, e])_{\frac{\delta\mathcal{I}}{\delta G^T} = \frac{\delta\mathcal{I}}{\delta e} = 0} = -\frac{1}{\pi} \int d\mathbb{X} \int d^2 z \, \mathrm{Tr}[(\partial_z G) G^{-1} (\partial_{\bar{z}} G) G^{-1}]. \tag{140}$$

To proceed, we employ the Polyakov-Wiegmann formula to establish the following identity:

$$\mathcal{S}[G^T G] = 2\mathcal{S}[G] - \frac{1}{\pi} \int d^2 z \, \mathrm{Tr}[G^{T-1}(\partial_{\bar{z}} G^T)(\partial_z G) G^{-1}] = 0. \tag{141}$$

From this result, we deduce that:

$$(\Theta[G, H, e])_{\frac{\delta\mathcal{I}}{\delta G} = \frac{\delta\mathcal{I}}{\delta e} = 0} = 2 \int d\mathbb{X} \, \mathcal{S}[G]. \tag{142}$$

Substituting this expression into Eq. (129), we obtain:

$$\mathcal{W}_{AdS_3} := (\mathcal{I}[G, H, e])_{\frac{\delta\mathcal{I}}{\delta G} = \frac{\delta\mathcal{I}}{\delta e} = 0} = \int d\mathbb{X} \, \mathcal{S}[G]. \tag{143}$$

Finally, by substituting the holomorphic potential density, $h^a_{\sigma,\Delta}$, as derived in Eq. (127), into the on-shell effective action, we recover the generating functional for the leaf amplitudes, thereby completing the proof.

---

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
