# Peer review of "An $AdS_{3}$ Dual for Supersymmetric MHV Celestial Amplitudes"

_SciPost Physics_

## Round 1 · Referee Report · Anonymous (Referee 1) · 2025-10-29

Strengths

  • claims to give a construction which provides a solution to an important problem in celestial holography

Weaknesses

  • difficult for the reader to parse the details of the construction without an unreasonable level of background knowledge in a large number of niche works

Report

The main result of this paper is a holographic construction of the graviton MHV amplitude in N=8 supergravity, which is of relevance to the celestial holography program. The problem that the author is claiming to address is indeed an important and ambitious one, and is of direct relevance to a large number of active researchers, so I recommend it for publication given some major revisions. The revisions ask for clarity on a number of points, as well as corrections to equations which may contain mistakes.

I will say that this paper was difficult to review for many reasons, but mainly because it utilizes many pre-existing constructions from older works in order to make its claims. Some of these older works are by the author himself, and some are very niche works by other authors from many years ago.

There are certainly very few readers who will be familiar with all of the many references and constructions necessary to understand this paper. However, while these past works are indeed cited in this present paper, it unfortunately reads as a bit of a ''soup'' and it is difficult to ''get to the bottom'' of all of the definitions to disentangle what is new and what is used from previous works. In the future, I hope the author organizes his papers in a way which is easier for outsiders to penetrate, as they seem interesting and ambitious.

The following points outline the revisions necessary in order to be published.

Requested changes

1- In equation 8, reproduced below, an operator is defined as \begin{equation} \mathcal{P}i := \frac{1}{i} \frac{\bar{\nu}^{\dot A}_i \lambda^A}{\langle \nu_i, \lambda \rangle} \frac{\partial}{\partial X^{A \dot A}} e^{i p_i \cdot X} \end{equation} with $p_i^{A \dot A} = \nu_i^A \bar{\nu}_i^{\dot A}$. It seems as though \begin{equation} \frac{\partial}{\partial X^{A \dot A}} e^{i p_i \cdot X} = i \, p \end{equation} but this would imply \begin{equation} \mathcal{P}_i = 0 \end{equation} because } e^{i p_i \cdot X} = i \, \nu_{i A} \bar{\nu}_{i \dot A} e^{i p_i \cdot X$\bar{\nu}i^{\dot A} \bar{\nu} = 0$. So why is this operator not zero?

2- On page 6, the author introduces an auxiliary reference spinor $\lambda$. The first equation the reference spinor $\lambda$ appears in is equation (8), where it is used to define $\mathcal{P}_i$.

Furthermore, the only ''state'' that has been define up until this point is the fermionic vacuum state $| 0 \rangle$.

However, a new state appears in equation (9)/(10). (Actually, as equations (9) and (10) are really the same equation, the author should use the "aligned" environment for multiline equations, but this is an irrelevant aside.)

Namely, in (9)/(10), the author writes down a state that seems to depend on this reference spinor, namely $| \lambda \rangle$. My confusion is that it is not clear how this state $| \lambda \rangle$ is defined. My best guess is that it is \textit{not} the state that should depend on $\lambda$, but really the \textit{operator} $\mathcal{P}_i$, which we could denote $\mathcal{P}_i^\lambda$. The author must explain what they means by $|\lambda \rangle$.

3- A small typo: in equation (28) and underneath the author uses $\tilde{\phi}_{\alpha \beta}$ and $\tilde{\varphi}_{\alpha \beta}$ to represent the same variable.

4- As far as I am aware, equation (47) is not correct. The author may have incorrectly assumed this result to hold in a large $N$ limit from a well known 1988 paper by Nair, ''A Current Algebra for Some Gauge Theory Amplitudes.'' In this paper, Nair claims to reproduce the Parke-Taylor gluon MHV formula from a level $k=1$ WZW model. However, Nair drops all multi-trace (higher order in $k$) terms, and it is unclear from the paper if he is aware he is dropping them. The history was reviewed nicely in a recent paper 2509.12200. In any case, the author seems to be making the same mistake that Nair did in dropping the multi trace terms.

I suspect the author most likely thinks that this equation holds in the large $N$ limit, with some $1/N$ corrections, but I do not think this is true. All the multitrace terms are of the same order in $N$ as the single trace terms (order $\mathcal{O}(1)$), unless I am mistaken. Unless the author can explain to me (simply) why (47) holds in the large $N$ limit, it seems to me to be a substantial mistake which needs to be remedied. At the very least, the author should make it clear that the multi-trace terms are to be dropped.

A previous work by the author, 2408.10944, cites a book by Ketov for this fact, but I was unable to locate this fact in that book.

5- One of the most interesting claims of the paper is that it claims to give a generating function for the MHV amplitude, in section IIC. In this section, the author cites some notes by Nair, reference [33], which do not seem to be available on the internet. As such it is not clear to me if this generating function comes from the author or Nair. In any case, the claim, summarized in equation (59), is nonetheless quite interesting.

It would be helpful if the author could give an example of how (59) can be used to compute an amplitude, maybe just the 3-point amplitude for simplicity.

Recommendation

Ask for major revision

---

## Editorial Decision

in_refereeing